# Mean Flow Policy Optimization

**Xiaoyi Dong** [1 2]   **Xi Sheryl Zhang** [1]   **Jian Cheng** [1 3 4]

## Abstract

Diffusion models have recently emerged as expressive policy representations for online reinforcement learning (RL). However, their iterative generative processes introduce substantial training and inference overhead. To overcome this limitation, we propose to represent policies using MeanFlow models, a class of few-step flow-based generative models, to improve training and inference efficiency over diffusion-based RL approaches. To promote exploration, we optimize MeanFlow policies under the maximum entropy RL framework via soft policy iteration, and address two key challenges specific to MeanFlow policies: action likelihood evaluation and soft policy improvement. Experiments on MuJoCo, DeepMind Control Suite and HumanoidBench benchmarks demonstrate that our method, Mean Flow Policy Optimization (MFPO), achieves performance comparable to or exceeding current diffusion-based baselines while considerably reducing training and inference time. Our code is available at https://github.com/dongxiaoyi-xyz/MFPO.

## 1. Introduction

Online reinforcement learning (RL), where agents learn decision-making policies through interactions with an environment, has proven to be a powerful framework for solving continuous control tasks (Lillicrap et al., 2015; Schulman et al., 2015; 2017; Haarnoja et al., 2018; Fujimoto et al., 2018). Most existing approaches parameterize the policy as a mapping from states to either deterministic actions or Gaussian distributions over the action space. Although such policy classes are theoretically sufficient to represent optimal policies—one of which must be deterministic—their limited expressiveness often leads to inefficient exploration, thus prohibiting policy optimization. Specifically, in complex control tasks, the reward landscape is typically multi-modal. However, Gaussian policies and deterministic policies augmented with Gaussian noise are inherently unimodal, restricting exploration to a single action region at each state. As a result, other potentially high-reward regions may be overlooked, causing the agent to become trapped in suboptimal local optima.

Recent works have introduced diffusion and flow models as policy representations to address this limitation (Yang et al., 2023; Wang et al., 2024; Ding et al., 2024). By generating actions through an iterative process that progressively transforms noise into actions, these generative policies can model highly complex and multi-modal action distributions, leading to improved exploration and strong empirical performance in online RL. However, their practical applicability is hindered by the high computational cost of multi-step iterative generation, which substantially increases training and inference time compared to conventional one-step policies.

To mitigate this efficiency bottleneck, we adopt the recently proposed MeanFlow models (Geng et al., 2025) as policy representations. By shifting the learning target from the instantaneous velocity field to the average velocity along the sampling trajectory, MeanFlow models significantly reduce discretization error under coarse time discretization, enabling high-quality generation with only a few sampling steps. As a result, few-step MeanFlow policies retain the ability to model complex multi-modal action distributions, thereby facilitating efficient exploration across multiple promising action regions, while significantly reducing computational overhead in diffusion-based RL methods.

Furthermore, following the maximum entropy (MaxEnt) RL paradigm, we incorporate policy entropy into the optimization objective to encourage sufficient exploration of the state–action space, and adopt soft policy iteration to optimize MeanFlow policies. However, similar to other diffusion-based methods (Celik et al., 2025; Dong et al., 2025; Ma et al., 2025), applying soft policy iteration to MeanFlow policies introduces two major challenges. First, evaluating the soft Q-function requires computing the action likelihood of a MeanFlow policy, whose exact expression

---

[1]C²DL, Institute of Automation, Chinese Academy of Sciences
[2]School of Artificial Intelligence, University of Chinese Academy of Sciences [3]AiRiA [4]School of Future Technology, University of Chinese Academy of Sciences. Correspondence to: Jian Cheng <jcheng@nlpr.ia.ac.cn>.

*Proceedings of the 43$^{rd}$ International Conference on Machine Learning*, Seoul, South Korea. PMLR 306, 2026. Copyright 2026 by the author(s).

involves an intractable integral over the divergence of the instantaneous velocity field. Second, standard MeanFlow training relies on samples from the target distribution, which are generally unavailable in RL settings.

To address these challenges, we develop an *average divergence network* to approximate the intractable integral appearing in the action likelihood, and introduce an *adaptive instantaneous velocity estimation* method to construct a tractable training objective for MeanFlow policies. The resulting algorithm, termed Mean Flow Policy Optimization (MFPO), enables efficient and principled integration of MeanFlow models into the MaxEnt RL framework. We evaluate MFPO on benchmark tasks from MuJoCo and DeepMind Control Suite. Experimental results demonstrate that MFPO matches or surpasses the performance of existing diffusion-based RL algorithms, while requiring significantly fewer sampling steps and substantially less training and inference time.

## 2. Related Work

**Diffusion and Flow Models.** Diffusion models (Sohl-Dickstein et al., 2015) have become a dominant paradigm in generative modeling due to their training stability and strong empirical performance (Dhariwal & Nichol, 2021; Saharia et al., 2022; Brooks et al., 2024). These models gradually transform data distribution into a tractable prior distribution by adding Gaussian noise, and then train a neural network to reverse this process for sample generation. The reverse process can be formulated as a sequence of Gaussian transitions, a stochastic differential equation (SDE), or an equivalent ordinary differential equation (ODE) (Ho et al., 2020; Song & Ermon, 2019; Song et al., 2021). Flow Matching methods further extend this framework by directly modeling the velocity field of the corresponding ODE (Liu et al., 2022; Lipman et al., 2023). Since diffusion-based and flow-based methods share similar formulations and practical implementations, we collectively refer to them as diffusion models.

Despite the powerful generative capabilities, diffusion models are constrained by their long inference time due to the iterative sampling process. Therefore, a line of research focuses on reducing sampling steps without sacrificing performance. Distillation-based approaches (Salimans & Ho, 2022; Sauer et al., 2024) train a few-step student model to mimic the outputs of a pretrained multi-step teacher model. DPM-Solver (Lu et al., 2022; 2025) designs specialized high-order ODE solvers by exploiting the structure of diffusion ODEs. Consistency Models (Song et al., 2023; Song & Dhariwal, 2024) learn to directly map points along an ODE trajectory to its start point and enforce consistency among predictions from different time points during training. Shortcut Models (Frans et al., 2025) incorporate the step size as an additional condition and impose a self-consistency objective to ensure that models with larger step sizes behave consistently with those using smaller step sizes.

More recently, instead of learning instantaneous velocity fields as in standard flow models, MeanFlow models (Geng et al., 2025) propose to model the average velocity field, which is defined as the time integral of the instantaneous velocity divided by the corresponding time interval. When the average velocity field is learned accurately, samples can be generated without any discretization error even using very few sampling steps. MeanFlow models achieve superior sample quality compared to competing methods and can be trained from scratch without pretraining, distillation, or curriculum learning. Consequently, we adopt MeanFlow Models as policy representations to strike an effective balance among expressivity, inference speed, and simplicity.

**Diffusion Policies for Online RL.** There is a growing trend toward employing diffusion policies in online RL. In addition to diffusion policy optimization, a key challenge in this setting is to effectively balance exploration and exploitation. DIPO (Yang et al., 2023) clones the actions stored in the replay buffer and improves them by applying Q-function gradients. QVPO (Ding et al., 2024) and FPMD (Chen et al., 2025) weights the diffusion/flow loss on generated actions with the Q-function for policy improvement, and QVPO introduces an additional diffusion loss on uniformly sampled actions as a regularization term to encourage exploration. GenPO (Ding et al., 2025) constructs an invertible diffusion model whose log-likelihood can be computed exactly via the change-of-variable formula, enabling compatibility with classical on-policy RL algorithms such as PPO. FlowRL (Lv et al., 2025) combines the Q-loss and the diffusion loss weighted by the optimal Q-function, aiming to optimize policies within the neighborhood of the optimal behavior policy. QSM (Psenka et al., 2024) uses the gradient of the Q-function as the regression target for the score function. DACER (Wang et al., 2024) backpropagates Q-function gradients through the entire diffusion chain to optimize the policy parameters. Both QSM and DACER rely on injecting additional Gaussian noise into the generated actions to promote exploration.

Rather than heuristically exploring near the current policy by adding Gaussian noise, maximum entropy reinforcement learning provides a principled framework for balancing exploration and exploitation. However, optimizing diffusion policies under a maximum entropy objective is non-trivial, as both entropy computation and soft policy improvement are intractable for diffusion-based policies. DIME (Celik et al., 2025) derives a lower bound on the policy entropy and updates the policy by backpropagating gradients through the diffusion chain. To approximate the Boltzmann policy induced by the Q-function, MaxEntDP and SDAC (Dong et al., 2025; Ma et al., 2025) sample clean actions from a Gaussian

distribution conditioned on noisy actions and weight the diffusion loss using the Q-function. RFM (Li et al., 2026) constructs the policy training objective using a class of posterior mean estimators, and introduces control variates based on Langevin Stein operators to reduce estimator variance. In addition, MaxEntDP estimates the policy entropy via computationally expensive numerical integration. However, these methods degrade when only a few sampling steps are used. Specifically, the entropy lower bound employed by DIME becomes loose under small step budgets, which can adversely affect policy learning. MaxEntDP, SDAC and RFM suffer from significant discretization error when the diffusion ODE is solved using only a limited number of sampling steps. Recently, SAC-Flow (Zhang et al., 2026) parameterizes diffusion policies using sequence architectures, such as GRUs or Transformers, and optimizes them within the Soft Actor-Critic framework (Haarnoja et al., 2018). Although this design improves training stability, it relies on specialized architectural choices, thereby increasing implementation complexity.

Currently, most diffusion-based online RL methods require 10–20 sampling steps to achieve strong empirical performance, leading to substantially slower training and inference compared to approaches based on conventional one-step policies. In this work, we investigate the use of few-step MeanFlow policies to improve both training and inference efficiency, and optimize them under the maximum entropy framework to achieve efficient exploration.

## 3. Preliminary

### 3.1. Maximum Entropy Reinforcement Learning

We consider policy learning in continuous action spaces. The environment is modeled as a Markov Decision Process (MDP) defined by the tuple $(\mathcal{S}, \mathcal{A}, p, r, \rho_0, \gamma)$, where $\mathcal{S}$ and $\mathcal{A}$ denote the state space and the action space, respectively. The transition dynamics $p : \mathcal{S} \times \mathcal{S} \times \mathcal{A} \to \mathbb{R}^+$ defines the probability density function of the next state $\boldsymbol{s}_{t+1} \in \mathcal{S}$ given the current state $\boldsymbol{s}_t \in \mathcal{S}$ and the action $\boldsymbol{a}_t \in \mathcal{A}$, $r : \mathcal{S} \times \mathcal{A} \to [r_{\min}, r_{\max}]$ is a bounded reward function, $\rho_0$ denotes the initial state distribution, and $\gamma \in [0, 1]$ is the discount factor. Given a policy $\pi(\boldsymbol{a}_t | \boldsymbol{s}_t)$, we denote the marginal state-action distribution induced by $\pi$ as $\rho_\pi(\boldsymbol{s}_t, \boldsymbol{a}_t)$. And for notational convenience, the reward $r(\boldsymbol{s}_t, \boldsymbol{a}_t)$ at time $t$ is abbreviated as $r_t$.

Maximum Entropy (MaxEnt) RL augments the standard RL objective by incorporating policy entropy to encourage learning stochastic policies. The resulting objective is

$$J(\pi) = \sum_{t=0}^{\infty} \gamma^t \mathbb{E}_{\rho_\pi} \left[ r_t + \alpha \mathcal{H}(\pi(\cdot | \boldsymbol{s}_t)) \right], \quad (1)$$

where $\mathcal{H}(\pi(\cdot | \boldsymbol{s}_t)) = \mathbb{E}_{\boldsymbol{a}_t \sim \pi(\cdot | \boldsymbol{s}_t)} [-\log \pi(\boldsymbol{a}_t | \boldsymbol{s}_t)]$ is the pol-

icy entropy and $\alpha \in \mathbb{R}^+$ is a temperature parameter that controls the trade-off between reward maximization and entropy regularization. Under this objective, we are concerned with the soft Q function of a policy $\pi$, which is defined as

$$Q^\pi(\boldsymbol{s}_t, \boldsymbol{a}_t) = r_t + \sum_{l=1}^{\infty} \gamma^l \mathbb{E}_{\rho_\pi} \left[ r_{t+l} + \alpha \mathcal{H}(\pi(\cdot | \boldsymbol{s}_{t+l})) \right]. \quad (2)$$

### 3.2. Soft Policy Iteration

The MaxEnt RL objective can be achieved by applying soft policy iteration (Haarnoja et al., 2018). This algorithm alternates between policy evaluation and policy improvement, and can converge to the optimal MaxEnt policy under certain assumptions. In the policy evaluation step, the soft Q function of the current policy $\pi$ is learned by repeatedly applying the soft Bellman operator until convergence:

$$\mathcal{T}^\pi Q(\boldsymbol{s}_t, \boldsymbol{a}_t) \triangleq r_t + \gamma \mathbb{E}[Q(\boldsymbol{s}_{t+1}, \boldsymbol{a}_{t+1}) - \alpha \log \pi(\boldsymbol{a}_{t+1} | \boldsymbol{s}_{t+1})]. \quad (3)$$

In the policy improvement step, a new policy is obtained by projecting the Boltzmann distribution induced by the learned Q function onto the policy set $\Pi$ by minimizing the Kullback-Leibler divergence:

$$\pi_{\text{new}} = \arg\min_{\pi \in \Pi} \mathrm{D}_{\text{KL}} \left( \pi(\cdot | \boldsymbol{s}) \,\middle\|\, \frac{\exp\left(\frac{1}{\alpha} Q^{\pi_{\text{old}}}(\boldsymbol{s}, \cdot)\right)}{Z(\boldsymbol{s})} \right), \quad (4)$$

where $Z(\boldsymbol{s})$ denotes the partition function that normalizes the Boltzmann distribution. It can be proved that $Q^{\pi_{\text{new}}}(\boldsymbol{s}, \boldsymbol{a}) \geq Q^{\pi_{\text{old}}}(\boldsymbol{s}, \boldsymbol{a})$ for all $(\boldsymbol{s}, \boldsymbol{a}) \in \mathcal{S} \times \mathcal{A}$, indicating that the new policy improves upon the old one. By repeatedly alternating between policy evaluation and policy improvement, the algorithm will converge to the optimal MaxEnt policy in the policy set $\Pi$ (Haarnoja et al., 2018).

### 3.3. Flow Matching and Mean Flow Models

Flow Matching is a class of generative models that learns a continuous flow between two probability distributions. Formally, let $p_1(\boldsymbol{x}_1)$ denote a simple prior distribution (e.g., a standard Gaussian) and $p_0(\boldsymbol{x}_0)$ denote the data distribution. Flow matching aims to learn a time-dependent velocity field $\boldsymbol{v}(\boldsymbol{x}, t) : \mathbb{R}^d \times [0, 1] \to \mathbb{R}^d$, which defines the ODE

$$\frac{\mathrm{d}\boldsymbol{x}_t}{\mathrm{d}t} = \boldsymbol{v}(\boldsymbol{x}_t, t).$$

The induced flow should transport the prior distribution $p_1$ at $t = 1$ to the data distribution $p_0$ at $t = 0$. A feasible flow path can be constructed as $\boldsymbol{x}_t = a_t \boldsymbol{x}_0 + b_t \boldsymbol{x}_1$, where $a_t$ and $b_t$ are predefined interpolation schedules. In this paper, we adopt the commonly used linear schedule $a_t = 1 - t$ and $b_t = t$. Accordingly, the conditional velocity field is given by $\boldsymbol{v}_t(\boldsymbol{x}_t | \boldsymbol{x}_0) = \frac{\mathrm{d}\boldsymbol{x}_t}{\mathrm{d}t} = \boldsymbol{x}_1 - \boldsymbol{x}_0$. A parameterized neural network $\boldsymbol{v}_\theta$ is then trained to approximate the marginal velocity field by marginalizing over the conditional velocity. The resulting Conditional Flow Matching (CFM) loss is

$$\mathcal{L}_{\text{CFM}}(\theta) = \mathbb{E}_{t, \boldsymbol{x}_0, \boldsymbol{x}_1} \left\| \boldsymbol{v}_\theta(\boldsymbol{x}_t, t) - \boldsymbol{v}_t(\boldsymbol{x}_t | \boldsymbol{x}_0) \right\|^2. \quad (5)$$

After training, new samples are generated by first sampling $\boldsymbol{x}_1 \sim p_1$ and then solving the learned ODE backward in time from $t = 1$ to $t = 0$. The solution can be written as $\boldsymbol{x}_0 = \boldsymbol{x}_1 - \int_0^1 \boldsymbol{v}(\boldsymbol{x}_t, t)\, \mathrm{d}t$. In practice, a first-order Euler solver approximates this integral using $T$ discrete steps:

$$\boldsymbol{x}_{t_{i-1}} = \boldsymbol{x}_{t_i} - (t_i - t_{i-1})\boldsymbol{v}(\boldsymbol{x}_{t_i}, t_i),$$

where $\{t_i\}_{i=0}^T$ is an increasing sequence in $[0, 1]$. Although effective with sufficiently many steps, the Euler method incurs large discretization errors when $T$ is small.

To address this limitation, MeanFlow Models (Geng et al., 2025) propose to directly model the *average velocity field*, defined as the ratio of the time-integrated instantaneous velocity to the time interval:

$$\boldsymbol{u}(\boldsymbol{x}_t, r, t) \triangleq \frac{1}{t - r} \int_r^t \boldsymbol{v}(\boldsymbol{x}_\tau, \tau)\, \mathrm{d}\tau. \qquad (6)$$

Based on this definition, the instantaneous and average velocity fields satisfy the *MeanFlow Identity*:

$$\boldsymbol{u}(\boldsymbol{x}_t, r, t) = \boldsymbol{v}(\boldsymbol{x}_t, t) - (t - r)\frac{\mathrm{d}}{\mathrm{d}t}\boldsymbol{u}(\boldsymbol{x}_t, r, t), \qquad (7)$$

where the time derivative is given by $\frac{\mathrm{d}}{\mathrm{d}t}\boldsymbol{u}(\boldsymbol{x}_t, r, t) = \boldsymbol{v}(\boldsymbol{x}_t, t)\partial_{\boldsymbol{x}}\boldsymbol{u} + \partial_t \boldsymbol{u}$. The right-hand side of Eq. (7) is used as the training target for the average velocity network $\boldsymbol{u}_\theta$. The training objective is

$$\mathcal{L}(\theta) = \mathbb{E}_{r, t, \boldsymbol{x}_t} \|\boldsymbol{u}_\theta(\boldsymbol{x}_t, r, t) - \mathrm{sg}(\boldsymbol{u}_{\mathrm{tgt}})\|_2^2, \qquad (8)$$

where $\boldsymbol{u}_{\mathrm{tgt}} = \boldsymbol{v}(\boldsymbol{x}_t, t) - (t - r)(\boldsymbol{v}(\boldsymbol{x}_t, t)\partial_{\boldsymbol{x}}\boldsymbol{u}_\theta + \partial_t \boldsymbol{u}_\theta)$, and $\mathrm{sg}(\cdot)$ denotes the stop-gradient operator. In practice, the marginal velocity filed can be replaced by its unbiased estimator, the conditional velocity field to compute the training target. Once trained, sampling can be performed by replacing the time integral with the average velocity:

$$\boldsymbol{x}_r = \boldsymbol{x}_t - (t - r)\boldsymbol{u}(\boldsymbol{x}_t, r, t).$$

Unlike the instantaneous velocity field, integrating the average velocity yields an accurate approximation of the trajectory integral regardless of the number of sampling steps.

## 4. Method

In this paper, we adopt MeanFlow models as the policy representation and employ soft policy iteration to optimize the MaxEnt RL objective. We first present the formulation of MeanFlow policies in Sec. 4.1. To address the challenges arising from their integration into soft policy iteration, we introduce an *average divergence network* to approximate the action likelihood of MeanFlow policies in Sec. 4.2, and propose an *adaptive instantaneous velocity estimation* method to enable effective policy improvement in Sec. 4.3. Finally, we describe a practical implementation of the proposed algorithm in Sec. 4.4.

### 4.1. Mean Flow Policy Representation

Formally, we represent the policy as the solution to the ODE

$$\frac{\mathrm{d}\boldsymbol{a}_t}{\mathrm{d}t} = \boldsymbol{v}(\boldsymbol{s}, \boldsymbol{a}_t, t). \qquad (9)$$

Starting from a prior sample $\boldsymbol{a}_1 \sim p_1$, the solution can be given by $\boldsymbol{a}_0 = \boldsymbol{a}_1 - \int_0^1 \boldsymbol{v}(\boldsymbol{s}, \boldsymbol{a}_t, t)dt$. Note in this paper, the variable $t$ denotes the ODE time rather than RL time index unless specified. $\boldsymbol{v}$ is called the instantaneous velocity filed and the corresponding average velocity filed is defined as

$$\boldsymbol{u}(\boldsymbol{s}, \boldsymbol{a}_t, r, t) = \frac{1}{t - r} \int_r^t \boldsymbol{v}(\boldsymbol{s}, \boldsymbol{a}_\tau, \tau)\, \mathrm{d}\tau. \qquad (10)$$

A neural network $\boldsymbol{u}_\theta$ parameterized by $\theta$ is trained to fit the average velocity filed. According to Eq. (10), a parameterized instantaneous velocity filed can be obtained by setting $r = t$: $\boldsymbol{v}_\theta(\boldsymbol{s}, \boldsymbol{a}_t, t) = \boldsymbol{u}_\theta(\boldsymbol{s}, \boldsymbol{a}_t, t, t)$. After training, actions are generated by accumulating the average velocity over a finite number of sampling steps, which serves as an approximation to the integral of the instantaneous velocity:

$$\boldsymbol{a}_{t_{i-1}} = \boldsymbol{a}_{t_i} - \frac{1}{T}\boldsymbol{u}_\theta(\boldsymbol{s}, \boldsymbol{a}_{t_i}, t_{i-1}, t_i), \quad i = T, \ldots, 1, \qquad (11)$$

where $T$ is the number of sampling steps and $t_i = \frac{i}{T}$. We refer to the induced action distribution conditioned on state $\boldsymbol{s}$ as the *MeanFlow policy* and denote it by $\pi_\theta(\boldsymbol{a}_0|\boldsymbol{s})$.

### 4.2. Training Average Divergence Network for Action Likelihood Approximation

Using the instantaneous change-of-variable formula (Chen et al., 2018), the action likelihood of a MeanFlow policy can be expressed as

$$\log \pi_\theta(\boldsymbol{a}_0|\boldsymbol{s}) = \log p_1(\boldsymbol{a}_1) + \int_0^1 \nabla \cdot \boldsymbol{v}_\theta(\boldsymbol{s}, \boldsymbol{a}_t, t)\, \mathrm{d}t. \qquad (12)$$

Here, $\nabla\cdot$ denotes the divergence operator with respect to the action variable, which can be written as the trace of the Jacobian of the instantaneous velocity field:

$$\nabla \cdot \boldsymbol{v}_\theta(\boldsymbol{s}, \boldsymbol{a}_t, t) = \mathrm{tr}\left(\frac{\partial \boldsymbol{v}_\theta(\boldsymbol{s}, \boldsymbol{a}_t, t)}{\partial \boldsymbol{a}_t}\right). \qquad (13)$$

A naive approach to computing the action likelihood is to explicitly evaluate the Jacobian in Eq. (13) and numerically integrate the divergence over time. However, both Jacobian computation and time integration are computationally expensive, which would substantially slow down soft policy iteration. Therefore, we seek a more efficient method for likelihood estimation.

Inspired by the concept of average velocity in MeanFlow models, we introduce an *average divergence network* $\delta_\omega$ to

approximate the time-averaged divergence:

$$\delta(\boldsymbol{s}, \boldsymbol{a}_t, r, t) = \frac{1}{t-r} \int_r^t \nabla \cdot \boldsymbol{v}_\theta(\boldsymbol{s}, \boldsymbol{a}_\tau, \tau) \, d\tau. \qquad (14)$$

This definition mirrors that of the average velocity field, with the instantaneous velocity replaced by the instantaneous divergence. According to MeanFlow theory, training the average divergence network requires an unbiased estimator of the instantaneous divergence.

Direct computation of the Jacobian trace requires $d$ backward passes through the network, where $d$ is the action dimension. Instead, we adopt the Skilling–Hutchinson trace estimator (Skilling, 1989; Hutchinson, 1989):

$$\boldsymbol{\epsilon}^\top \frac{\partial \boldsymbol{v}_\theta(\boldsymbol{s}, \boldsymbol{a}_t, t)}{\partial \boldsymbol{a}_t} \boldsymbol{\epsilon}, \qquad (15)$$

where $\boldsymbol{\epsilon}$ is sampled from a distribution satisfying $\mathbb{E}[\boldsymbol{\epsilon}] = \boldsymbol{0}$ and $\mathrm{Cov}(\boldsymbol{\epsilon}) = I$. This estimator is unbiased and can be efficiently computed using a single forward pass with forward-mode automatic differentiation. In practice, averaging multiple samples reduces variance and improves training stability. The resulting estimator is

$$\widehat{\mathrm{div}}(\boldsymbol{s}, \boldsymbol{a}_t, t, \boldsymbol{\epsilon}_{1:N}) = \frac{1}{N} \sum_{i=1}^N \boldsymbol{\epsilon}_i^\top \frac{\partial \boldsymbol{v}_\theta(\boldsymbol{s}, \boldsymbol{a}_t, t)}{\partial \boldsymbol{a}_t} \boldsymbol{\epsilon}_i, \qquad (16)$$

where $\{\boldsymbol{\epsilon}_i\}_{i=1}^N \sim p(\boldsymbol{\epsilon})$, and $N$ denotes the number of samples used for divergence estimation.

Following the construction of the MeanFlow loss, the training objective for the average divergence network is

$$\mathcal{L}(\omega) = \mathbb{E}_{\boldsymbol{s}, r, t, \boldsymbol{a}_t, \boldsymbol{\epsilon}_{1:N}} \left\| \delta_\omega(\boldsymbol{s}, \boldsymbol{a}_t, r, t) - \mathrm{sg}(\delta_{\mathrm{tgt}}) \right\|_2^2, \qquad (17)$$

where

$$\delta_{\mathrm{tgt}} = \widehat{\mathrm{div}} - (t-r) \left( \boldsymbol{u}_\theta(\boldsymbol{s}, \boldsymbol{a}_t, t, t) \, \partial_{\boldsymbol{a}} \delta_\omega + \partial_t \delta_\omega \right).$$

The full derivation is provided in the App. B.1. Once trained, the average divergence network can be incorporated into the MeanFlow sampling procedure to efficiently compute action likelihoods:

$$\log \pi_\theta(\boldsymbol{a}_0|\boldsymbol{s}) = \log p_1(\boldsymbol{a}_1) + \frac{1}{T} \sum_{i=1}^T \delta_\omega(\boldsymbol{s}, \boldsymbol{a}_{t_i}, t_{i-1}, t_i), \quad (18)$$

where $\{\boldsymbol{a}_{t_i}\}_{i=0}^T$ are samples generated using Eq. 11. The training procedure for the average divergence network and the likelihood-aware sampling process are summarized in Alg. 1 and 2, respectively. Empirically, we find that the proposed average divergence network provides accurate likelihood estimation while introducing only a 5% increase in training time.

### 4.3. Adaptive Instantaneous Velocity Estimation with Self-Normalized Importance Sampling

In the policy improvement step, the target distribution of the MeanFlow policy is the Boltzmann distribution induced by the soft Q-function:

$$\pi(\boldsymbol{a}_0|\boldsymbol{s}) = \frac{\exp\left(\frac{1}{\alpha} Q(\boldsymbol{s}, \boldsymbol{a}_0)\right)}{Z(\boldsymbol{s})}, \qquad (19)$$

where $Z(\boldsymbol{s})$ denotes the normalizing constant. In this subsection, we describe how to optimize the average velocity network to approximate this target distribution.

As shown in Eq. 8, training MeanFlow models only requires an unbiased estimator of the instantaneous velocity field of the target distribution. This velocity field can be obtained by marginalizing the conditional velocity field as

$$\boldsymbol{v}_t(\boldsymbol{a}_t|\boldsymbol{s}) = \mathbb{E}_{\pi(\boldsymbol{a}_0|\boldsymbol{a}_t, \boldsymbol{s})} \left[ \boldsymbol{v}_t(\boldsymbol{a}_t|\boldsymbol{a}_0, \boldsymbol{s}) \right] \qquad (20)$$

$$= \mathbb{E}_{\pi(\boldsymbol{a}_0, \boldsymbol{a}_1|\boldsymbol{a}_t, \boldsymbol{s})} \left[ \boldsymbol{a}_1 - \boldsymbol{a}_0 \right]. \qquad (21)$$

Standard MeanFlow training draws samples from the target distribution and compute the corresponding conditional velocity as a unbiased estimator of marginal velocity filed. However, in the policy improvement step, the target distribution is defined implicitly by the Q-function, and direct sampling is unavailable. To address this issue, we further analyze the marginal velocity field. Substituting the relation $\boldsymbol{a}_1 = \frac{\boldsymbol{a}_t - (1-t)\boldsymbol{a}_0}{t}$ into the above equation yields

$$\boldsymbol{v}_t(\boldsymbol{a}_t|\boldsymbol{s}) = \mathbb{E}_{\pi(\boldsymbol{a}_0|\boldsymbol{a}_t, \boldsymbol{s})} \left[ \frac{\boldsymbol{a}_t - \boldsymbol{a}_0}{t} \right]. \qquad (22)$$

MaxEntDP and SDAC (Dong et al., 2025; Ma et al., 2025) show that the conditional distribution associated with the Boltzmann policy admits the following form:

$$\pi(\boldsymbol{a}_0|\boldsymbol{a}_t, \boldsymbol{s}) \propto \exp\left(\tfrac{1}{\alpha} Q(\boldsymbol{s}, \boldsymbol{a}_0)\right) \mathcal{N}\left(\boldsymbol{a}_0 \,\Big|\, \tfrac{\boldsymbol{a}_t}{1-t}, \left(\tfrac{t}{1-t}\right)^2 I\right). \quad (23)$$

The complete derivation is provided in App. B.2. Let $q(\boldsymbol{a}_0) = \mathcal{N}\left(\boldsymbol{a}_0 | \frac{\boldsymbol{a}_t}{1-t}, \left(\frac{t}{1-t}\right)^2 I\right)$, similar to MaxEntDP, we can draw samples $\boldsymbol{a}_0 \sim q(\boldsymbol{a}_0)$ and estimate the marginal velocity field using self-normalized importance sampling (SNIS):

$$\boldsymbol{v}_t(\boldsymbol{a}_t|\boldsymbol{s}) = \mathbb{E}_{q(\boldsymbol{a}_0)} \left[ \frac{\pi(\boldsymbol{a}_0|\boldsymbol{a}_t, \boldsymbol{s})}{q(\boldsymbol{a}_0)} \cdot \frac{\boldsymbol{a}_t - \boldsymbol{a}_0}{t} \right] \qquad (24)$$

$$\approx \sum_{i=1}^K \frac{w(\boldsymbol{a}_0^i)}{\sum_{j=1}^K w(\boldsymbol{a}_0^j)} \cdot \frac{\boldsymbol{a}_t - \boldsymbol{a}_0^i}{t}, \qquad (25)$$

where the importance weight $w(\boldsymbol{a}_0) = \frac{\pi(\boldsymbol{a}_0|\boldsymbol{a}_t, \boldsymbol{s})}{q(\boldsymbol{a}_0)} \propto \exp\left(\frac{1}{\alpha} Q(\boldsymbol{s}, \boldsymbol{a}_0)\right)$, and $K$ denotes the number of samples.

However, this SNIS estimator is not efficient for all $t \in [0, 1]$. Its relative efficiency compared to a Monte Carlo estimator using samples from $\pi(\boldsymbol{a}_0|\boldsymbol{a}_t, \boldsymbol{s})$ can be quantified by the Effective Sample Size (ESS), which measures the number of Monte Carlo samples to ensure that the variance of the Monte Carlo estimator is the same as the SNIS estimator with $K$ samples (Kong, 1992; Metelli et al., 2020). An brief introduction to ESS is provided in App. B.3. Therefore, given the budget of $K$ samples, a SNIS estimator with a larger ESS has smaller estimation variance and is more efficient. The ESS of SNIS can be estimated as

$$\widehat{\text{ESS}}(\pi\|q) = \frac{\left(\sum_{i=1}^{K} w(\boldsymbol{a}_0^i)\right)^2}{\sum_{i=1}^{K} w^2(\boldsymbol{a}_0^i)}. \quad (26)$$

When the proposal distribution $q$ gradually deviates from the target distribution $\pi$, the ESS degrades from $K$ to 1. Therefore, a proposal distribution that is sufficiently close to the target distribution is desired. According to Eq. 23, when $t$ is close to zero, the target distribution is sharply concentrated due to the small variance of the Gaussian proposal, and the Gaussian proposal closely matches the target distribution. In contrast, when $t$ approaches one, the Gaussian proposal becomes much flatter and the target distribution is increasingly dominated by the exponential Q-function term, leading to a poor match between $q$ and the target distribution, and consequently, low ESS.

To improve estimation efficiency for large $t$, we introduce a second proposal distribution. This proposal should (i) be easy to sample from, (ii) admit tractable likelihood computation, and (iii) be close to the Boltzmann distribution induced by the Q-function which is neglected by the Gaussian proposal. With the likelihood estimation method developed in Sec.4.2, the current policy distribution $\pi_\theta(\boldsymbol{a}_0|\boldsymbol{s})$ satisfies all three requirements. As policy optimization proceeds, this distribution increasingly concentrates on high-Q regions, making it a suitable proposal. Moreover, to adaptively select the estimators based on their efficiency, we therefore combine two SNIS estimators using ESS as weights. Specifically, we define $q^1(\boldsymbol{a}_0) = \pi_\theta(\boldsymbol{a}_0|\boldsymbol{s})$. and $q^2(\boldsymbol{a}_0) = \mathcal{N}\left(\boldsymbol{a}_0|\frac{\boldsymbol{a}_t}{1-t}, (\frac{t}{1-t})^2 I\right)$. The final estimator of the marginal velocity field is

$$\hat{\boldsymbol{v}}_t(\boldsymbol{a}_t|\boldsymbol{s}) = \sum_{k=1}^{2} \frac{\text{ESS}_k}{\sum_{l=1}^{2} \text{ESS}_l} \hat{\boldsymbol{v}}_t^k(\boldsymbol{a}_t|\boldsymbol{s}), \quad (27)$$

where each $\hat{\boldsymbol{v}}_t^k$ is computed using Eq. 25 with importance weights $w^k(\boldsymbol{a}_0) = \frac{\pi(\boldsymbol{a}_0|\boldsymbol{a}_t, \boldsymbol{s})}{q^k(\boldsymbol{a}_0)}$. The normalizing constant of $\pi(\boldsymbol{a}_0|\boldsymbol{a}_t, \boldsymbol{s})$ cancels out in both SNIS and ESS computation and therefore does not need to be evaluated. A detailed bias and variance analysis of the proposed estimator is provided in App. B.4.

Finally, the training objective of the MeanFlow policy is given by

$$\mathcal{L}(\theta) = \mathbb{E}_{\boldsymbol{s}, r, t, \boldsymbol{a}_t} \|\boldsymbol{u}_\theta(\boldsymbol{s}, \boldsymbol{a}_t, r, t) - \text{sg}(\boldsymbol{u}_{\text{tgt}})\|_2^2, \quad (28)$$

where $\boldsymbol{u}_{\text{tgt}} = \hat{\boldsymbol{v}}_t - (t - r)(\hat{\boldsymbol{v}}_t \, \partial_{\boldsymbol{a}} \boldsymbol{u}_\theta + \partial_t \boldsymbol{u}_\theta)$.

In Fig. 1, we empirically evaluate the ESS and the estimation variance of the two proposal distributions across different values of $t$. It shows that the ESS of the Gaussian proposal $q^2$ drops greatly when $t$ becomes larger, while the policy proposal $q^1$ still has a relatively high ESS. By combining the two SNIS estimators, the estimation variance becomes less than that of any single estimator.

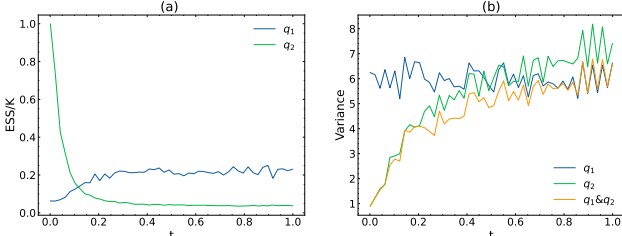

*Figure 1.* Effective sample size and estimation variance of two SNIS estimators under different proposals, evaluated on the HalfCheetah-v3 benchmark after 120k training iterations. (a) Normalized effective sample size (ESS divided by the number of samples), where larger values indicate more efficient proposals. (b) Estimation variance of the estimators, computed using the same number of samples.

### 4.4. A Practical Implementation

In this subsection, we present a practical implementation of the proposed Mean Flow Policy Optimization (MFPO) algorithm. We employ two parameterized neural networks: $Q_\phi$ to approximate the Q function and $\boldsymbol{u}_\theta$ to represent the MeanFlow policy. The algorithm follows the soft policy iteration framework, alternating between policy evaluation and policy improvement to iteratively optimize the MeanFlow policy.

In the policy evaluation step, the Q-function is trained by minimizing the soft Bellman error

$$\mathcal{L}(\phi) = \mathbb{E}_{(\boldsymbol{s}, \boldsymbol{a}, \boldsymbol{s}') \sim \mathcal{B}} \left[ \frac{1}{2} \left( Q_\phi(\boldsymbol{s}, \boldsymbol{a}) - Q_{\text{target}}(\boldsymbol{s}, \boldsymbol{a}) \right)^2 \right], \quad (29)$$

where $\mathcal{B}$ denotes the replay buffer that stores past interactions, and the target Q value is defined as $Q_{\text{target}}(\boldsymbol{s}, \boldsymbol{a}) = r(\boldsymbol{s}, \boldsymbol{a}) + \gamma \left( Q_\phi(\boldsymbol{s}', \boldsymbol{a}') - \alpha \log \pi_\theta(\boldsymbol{a}'|\boldsymbol{s}') \right)$. When computing the target Q value, the actions in the next timestep and their corresponding likelihoods are obtained according to Alg. 2.

In the policy improvement step, we optimize the MeanFlow policy using the loss defined in Eq. (28), which encourages the policy to approximate the Boltzmann distribution induced by the learned Q-function. Although the minimizer

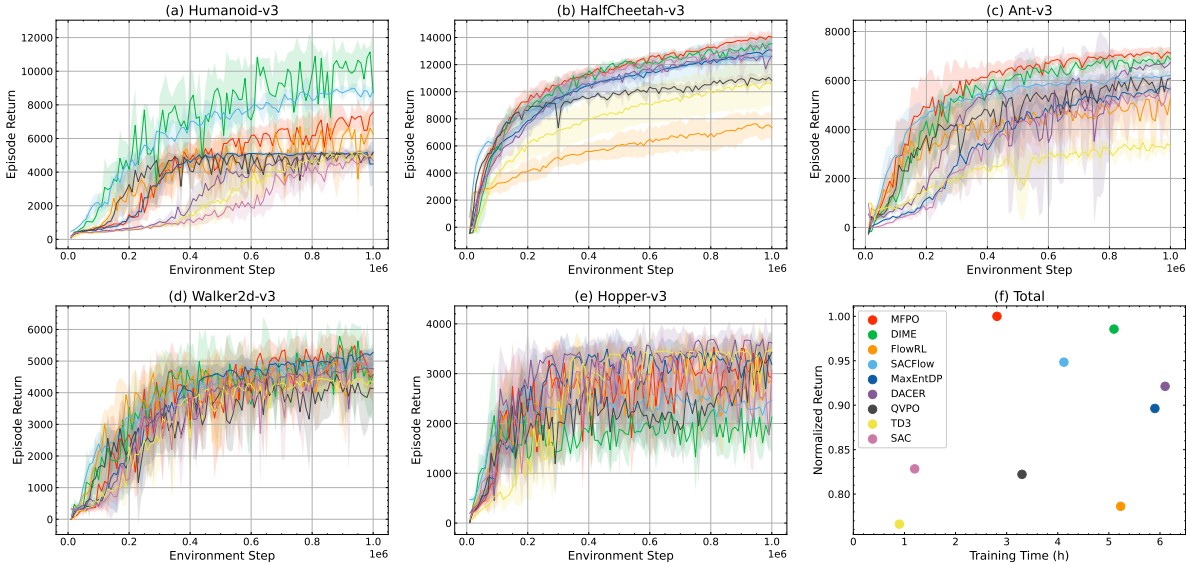

*Figure 2.* Comparison of different algorithms on 5 MuJoCo locomotion benchmarks. (a–e) Learning curves averaged over 5 random seeds; the shaded regions denote the standard deviation. (f) Average normalized return versus training time across all tasks. The normalized return on each task is computed by dividing the average return over the last 10% of environment steps by that of MFPO. Methods closer to the upper-left region exhibit both higher performance and greater time efficiency.

*Table 1.* Sampling steps and average inference latency per sample of different algorithms, evaluated on Mujoco benchmarks.

| Algorithm | MFPO | DIME | FlowRL | SAC-Flow | MaxEntDP | DACER | QVPO | TD3 | SAC |
|---|---|---|---|---|---|---|---|---|---|
| Sampling Steps | 2 | 16 | 1[1] | 4 | 20 | 20 | 20 | 1 | 1 |
| Inference Time (ms) | 0.46 | 0.97 | 0.42 | 0.96 | 1.56 | 1.06 | 1.68 | 0.14 | 0.15 |

of this objective generally does not coincide with that of the KL divergence in Eq. (4) when the Boltzmann distribution lies outside the policy class $\Pi$, we assume that the solutions are close in practice due to the expressiveness of the Mean-Flow policy class. After updating the policy network, the average diverge network is trained using the objective in Eq. (17). The complete MFPO algorithm is summarized in Alg. 3.

In addition, to further enhance the performance and stability of MFPO, we incorporate the following techniques.

**Auto-tuning Temperature.** The temperature coefficient $\alpha$ is a crucial hyperparameter that controls the trade-off between exploration and exploitation. Following SAC (Haarnoja et al., 2018), we automatically tune $\alpha$ to match a predefined target policy entropy $\mathcal{H}_{\text{target}}$, thereby alleviating the need for manual tuning. Specifically, $\alpha$ is optimized by minimizing

$$\mathcal{L}(\alpha) = \alpha\big(\mathcal{H}(\pi_\theta) - \mathcal{H}_{\text{target}}\big), \qquad (30)$$

where the policy entropy $\mathcal{H}(\pi_\theta)$ is estimated by averaging the negative log-likelihood of actions generated by the MeanFlow policy.

**Distributional Critic.** By explicitly modeling the distribu-

tion of returns, distributional Q-learning (Bellemare et al., 2017; Duan et al., 2021) has been shown to yield more stable and accurate policy evaluation. It has also demonstrated effectiveness in prior diffusion-based methods (Wang et al., 2024; Celik et al., 2025). Following C51 (Bellemare et al., 2017), we represent the Q-function as a categorical distribution over a discrete set of return values, and use the mean of this distribution as the final scalar Q value for policy optimization.

**Action Selection.** While stochastic policies are beneficial for exploration during training, they may underperform the deterministic policies at test time. To address this issue, we adopt the action selection technique (Mao et al., 2024; Ding et al., 2024; Espinosa-Dice et al., 2025). During evaluation, the MeanFlow policy generates multiple candidate actions, and the action with the highest estimated Q value is selected to interact with the environment.

## 5. Experiments

In this section, we evaluate the performance of the proposed MFPO approach. We conduct experiments on 5 locomotion

---

[1]FlowRL adopts a midpoint ODE solver; thus, one sampling step requires two forward network evaluations.

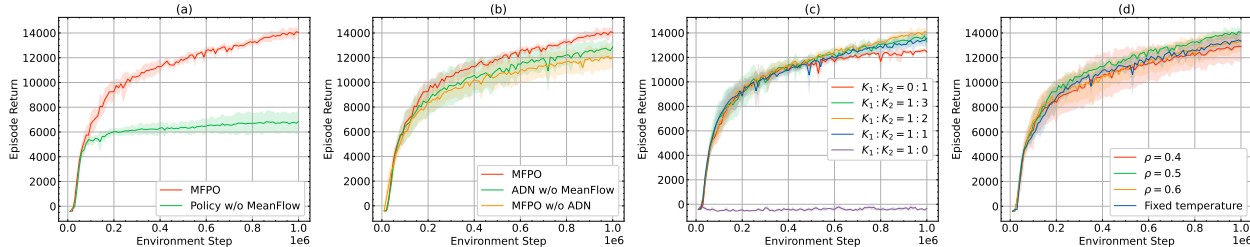

*Figure 3.* Ablation studies on the HalfCheetah-v3 benchmark. (a) Learning curves comparing policies that model average velocity versus instantaneous velocity. (b) Learning curves for action likelihood estimation using the average divergence network (ADN), the instantaneous divergence network, and no divergence network. (c) Learning curves under different sampling ratios of the two proposals in adaptive instantaneous velocity estimation. (d) Learning curves with different target entropy coefficients $\rho$ and a fixed temperature. The fixed temperature is set to the average temperature over the training process obtained with $\rho = 0.5$.

tasks from MuJoCo (Todorov et al., 2012), 6 hard tasks from DeepMind Control Suite (Tassa et al., 2018), and 3 high-dimensional tasks from HumanoidBench (Sferrazza et al., 2024). Comparisons with state-of-the-art baseline algorithms are presented in Sec. 5.1, and ablation studies analyzing the key components of MFPO are reported in Sec. 5.2.

## 5.1. Comparative Evaluation

To demonstrate the superiority of our approach, we compare MFPO with 6 diffusion-based algorithms—DIME (Celik et al., 2025), FlowRL (Lv et al., 2025), SAC-Flow (Zhang et al., 2026), MaxEntDP (Dong et al., 2025), DACER (Wang et al., 2024), and QVPO (Ding et al., 2024)—as well as 2 classical baselines employing Gaussian and deterministic policies, SAC (Haarnoja et al., 2018) and TD3 (Fujimoto et al., 2018). Performance and training time are reported in Fig. 2, Fig. 7 and Fig. 8, while the sampling steps and inference time are provided in Table 1. The results show that MFPO achieves comparable or superior performance to competing methods on most tasks. Notably, MFPO reduces training time by approximately 50% compared to diffusion-based policies, substantially narrowing the efficiency gap between diffusion policies and conventional one-step policies.

## 5.2. Ablation Studies

**MeanFlow Models.** MFPO employs MeanFlow models for both policy representation and divergence integral in action likelihood estimation. To assess the benefit of modeling the average velocity field, we replace these MeanFlow models with standard flow-matching models. As shown in Fig. 3(a) and (b), substituting the MeanFlow models in either the policy or the divergence integral leads to a noticeable performance degradation. These results highlight the importance of learning the average velocity in few-step settings, where it effectively mitigates discretization errors.

**Action Likelihood Estimation.** Both policy entropy eval-

uation and importance weight computation for the policy proposal depend on accurate action likelihood estimation. To examine the effect of the average divergence network (ADN), we remove its contribution by setting the divergence integral in the action likelihood expression to zero. The corresponding results are presented in Fig. 3(b). The MFPO variant without ADN underperforms the full model, demonstrating that precise action likelihood estimation enabled by ADN is critical to stable and effective policy optimization.

**Adaptive Instantaneous Velocity Estimation.** We investigate the impact of the sample ratio between the two proposals used in adaptive instantaneous velocity estimation, with results shown in Fig. 3(c). Combining the SNIS estimators from the policy proposal and the Gaussian proposal yields a more stable estimate of the instantaneous velocity, leading to improved performance compared to using only the Gaussian proposal. In contrast, relying solely on the policy proposal fails to produce a viable policy since this proposal becomes effective only when the current policy can already generate high-Q actions, which is difficult without assistance from the Gaussian proposal. Furthermore, we observe that allocating slightly more samples to the Gaussian proposal ($K_1 : K_2 = 1 : 2$) results in better performance. This may be attributed to the increased sample budget enabling the policy proposal to become effective more rapidly.

**Auto-tuning Temperature.** We set the target policy entropy proportional to the action-space dimension as $\mathcal{H}_{\text{target}} = -\rho \cdot \dim(\mathcal{A})$, where $\rho$ is a task-independent constant controlling the trade-off between exploration and exploitation. The performance under different values of $\rho$ as well as with a fixed temperature, is shown in Fig. 3(d). Too high $\rho$ may suppress exploration of potential high-return action regions, and too low $\rho$ may compromise reward maximization. Empirically, we find $\rho = 0.5$ provides a robust choice across most tasks. Moreover, compared to using a fixed temperature, which may suffer from insufficient exploration as reward scales increase during training, automatic temperature tuning maintains a consistent exploration level throughout training, facilitating stable policy improvement.

More ablation studies and hyperparameter analysis are provided in App. E.7 and E.8.

## 6. Conclusion

In this paper, we propose MFPO, a method that employs MeanFlow models as policy representations within the MaxEnt RL framework. By adopting few-step MeanFlow policies, MFPO achieves high expressivity with fast inference, substantially improving the training and inference speed of diffusion-based RL algorithms. To address the optimization challenges of MeanFlow policies, we introduce an average divergence network to approximate action likelihoods and propose an adaptive instantaneous velocity estimation method to construct the policy loss. Extensive experiments show that MFPO matches or outperforms diffusion-based baselines while requiring significantly less training time, inference time, and fewer sampling steps.

**Limitations and Future Work.** Although MFPO requires fewer sampling steps than existing diffusion-based algorithms, it still requires two sampling steps to achieve high performance. Future work will explore more advanced policy optimization techniques or integrate more powerful generative models to reduce the sampling steps to one.

## Acknowledgements

This work was supported by the National Key Research and Development Program of China No. 2021ZD0201504 and the CAAI-Qbosan Fund 2025CAAI-QBOSAN-06.

## Impact Statement

Diffusion-based policies are now widely adopted in Vision–Language–Action (VLA) systems for robot control, but often suffer from high computational cost. By optimizing few-step MeanFlow policies, our method improves training and inference efficiency while retaining high performance. This efficiency enables faster and more responsive robotic control and reduces computational and energy demands. Consequently, it may facilitate real-world deployment and broaden access to advanced robotic systems.

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

# A. Supplementary Related Work

## A.1. Diffusion Policies for Offline and Offline2Online RL.

Beyond the scope of online RL, diffusion policies have also demonstrated remarkable efficacy in offline and offline-to-online RL contexts. We summarize these approaches here to provide a broader perspective on the field. In offline RL, early studies apply diffusion policies to approximate complex behavior policies, which may exhibit high multi-modality and skewness (Pearce et al., 2023; Chi et al., 2025). Subsequent works attempt to improve the performance of diffusion policies beyond the behavior policy. SfBC and IDQL (Chen et al., 2023; Hansen-Estruch et al., 2023) generate multiple candidate actions and select actions by sampling from a softmax distribution over their Q-values. EDP and QIPO (Kang et al., 2023; Zhang et al., 2025a) reweight the diffusion loss using transformations of the Q-function to emphasize actions with higher Q values. DQL, CPQL, CAC and FQL (Wang et al., 2023; Chen et al., 2024; Ding & Jin, 2024; Park et al., 2025) introduce an additional Q-loss evaluated on actions generated by the diffusion policy and jointly optimize it with the diffusion loss. CEP and DAC (Lu et al., 2023; Fang et al., 2025) incorporate Q-guidance terms into the score function of the diffusion policy, steering the sampling process toward action regions with higher Q-values. Following the offline-to-online RL paradigm, DPPO and ReFlow (Ren et al., 2025; Zhang et al., 2025b) further include online interactions to fine-tune diffusion policies pretrained on offline datasets. They formulate the reverse diffusion process as a Markov Decision Process (MDP) with a simple Gaussian policy and apply the PPO (Schulman et al., 2017) algorithm to solve it.

## A.2. An Analysis for the Optimization Objective of DIME When Using Few Sampling Steps

The DIME algorithm (Celik et al., 2025) derives a variational lower bound on the entropy of diffusion policies. Then the policy entropy in the MaxEnt objective is replaced with this bound to obtain a lower bound on the original optimization objective. Specifically, the entropy bound is given by

$$\mathcal{H}(\pi(\boldsymbol{a}_0|\boldsymbol{s})) \geq \mathbb{E}_{\overleftarrow{\pi}}\left[\log\frac{\overrightarrow{\pi}(\boldsymbol{a}_{1:T}|\boldsymbol{s},\boldsymbol{a}_0)}{\overleftarrow{\pi}(\boldsymbol{a}_{0:T}|\boldsymbol{s})}\right], \tag{31}$$

where $\overrightarrow{\pi}$ and $\overleftarrow{\pi}$ denote the forward diffusion and reverse generation processes, respectively, and $T$ is the number of sampling steps. Consider the gap between the true entropy and the variational bound:

$$\Delta = \mathcal{H}(\pi(\boldsymbol{a}_0|\boldsymbol{s})) - \mathbb{E}_{\overleftarrow{\pi}}\left[\log\frac{\overrightarrow{\pi}(\boldsymbol{a}_{1:T}|\boldsymbol{s},\boldsymbol{a}_0)}{\overleftarrow{\pi}(\boldsymbol{a}_{0:T}|\boldsymbol{s})}\right] \tag{32}$$

$$= \mathbb{E}_{\overleftarrow{\pi}}\left[D_{\mathrm{KL}}\big(\overleftarrow{\pi}(\boldsymbol{a}_{1:T}|\boldsymbol{s},\boldsymbol{a}_0) \,\|\, \overrightarrow{\pi}(\boldsymbol{a}_{1:T}|\boldsymbol{s},\boldsymbol{a}_0)\big)\right]. \tag{33}$$

Under the Markovian assumption of both processes, the joint KL divergence decomposes as

$$\Delta = \sum_{t=1}^{T} \mathbb{E}_{\overleftarrow{\pi}}\left[D_{\mathrm{KL}}\big(\overleftarrow{\pi}(\boldsymbol{a}_{t-1}|\boldsymbol{a}_t,\boldsymbol{s}) \,\|\, \overrightarrow{\pi}(\boldsymbol{a}_{t-1}|\boldsymbol{a}_t,\boldsymbol{s})\big)\right], \tag{34}$$

which is a sum of step-wise KL divergence between the parametric reverse transitions and their ground-truth counterparts. From diffusion theory, the true reverse transition converges to a Gaussian distribution only in the infinitesimal limit $T \to \infty$ (Sohl-Dickstein et al., 2015; Ho et al., 2020). For finite $T$, the true reverse transition generally deviates from a Gaussian distribution and cannot be exactly captured by a Gaussian approximation. Since the reverse transition is typically parameterized as a Gaussian, each KL term remains strictly positive, resulting in a non-negligible gap and a loose bound in the few-step regime.

We empirically evaluate DIME under different sampling steps (Fig. 4). Reducing the number of steps from 16 to 2 leads to a substantial performance degradation. In contrast, our MFPO method achieves high performance with $T = 2$, suggesting that the degradation of DIME is not caused by the limited expressivity of few-step diffusion policies, but by the looseness of the variational objective. This motivates the development of alternative algorithms that are better suited for few-step diffusion policy learning.

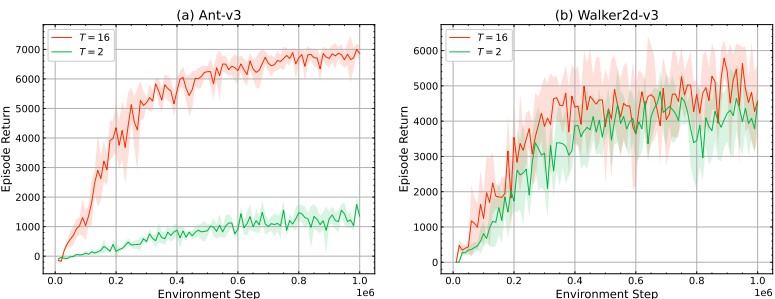

*Figure 4.* Learning curves of DIME algorithm using different sampling steps.

# B. Theoretical Derivations

## B.1. The Loss Derivation of the Average Divergence Network

The ground-truth average divergence is defined as

$$\delta(\boldsymbol{s}, \boldsymbol{a}_t, r, t) = \frac{1}{t - r} \int_r^t \nabla \cdot \boldsymbol{v}_\theta(\boldsymbol{s}, \boldsymbol{a}_\tau, \tau) \, \mathrm{d}\tau. \tag{35}$$

Multiplying both sides by $t - r$ and differentiating with respect to $t$ yield the identity

$$\delta(\boldsymbol{s}, \boldsymbol{a}_t, r, t) = \nabla \cdot \boldsymbol{v}_\theta(\boldsymbol{s}, \boldsymbol{a}_t, t) - (t - r) \frac{\mathrm{d}}{\mathrm{d}t} \delta(\boldsymbol{s}, \boldsymbol{a}_t, r, t), \tag{36}$$

where the total time derivative is given by

$$\frac{\mathrm{d}}{\mathrm{d}t} \delta(\boldsymbol{s}, \boldsymbol{a}_t, r, t) = \boldsymbol{v}_\theta(\boldsymbol{s}, \boldsymbol{a}_t, t) \, \partial_{\boldsymbol{a}} \delta + \partial_t \delta. \tag{37}$$

We take the right-hand side of Eq. (36) as the training target for the average divergence network $\delta_\omega$, leading to the objective

$$\mathcal{L}(\omega) = \mathbb{E}_{\boldsymbol{s}, r, t, \boldsymbol{a}_t} \left\| \delta_\omega(\boldsymbol{s}, \boldsymbol{a}_t, r, t) - \mathrm{sg}(\delta_{\mathrm{tgt}}) \right\|_2^2, \tag{38}$$

with

$$\delta_{\mathrm{tgt}} = \nabla \cdot \boldsymbol{v}_\theta(\boldsymbol{s}, \boldsymbol{a}_t, t) - (t - r) \left( \boldsymbol{v}_\theta(\boldsymbol{s}, \boldsymbol{a}_t, t) \, \partial_{\boldsymbol{a}} \delta + \partial_t \delta \right). \tag{39}$$

In practice, the divergence term is replaced by its unbiased estimator, and the instantaneous velocity is obtained from the average velocity network $\boldsymbol{u}_\theta(\boldsymbol{s}, \boldsymbol{a}_t, r, t)$ by setting $r = t$. The final training target therefore becomes

$$\delta_{\mathrm{tgt}} = \widehat{\mathrm{div}} - (t - r) \left( \boldsymbol{u}_\theta(\boldsymbol{s}, \boldsymbol{a}_t, t, t) \, \partial_{\boldsymbol{a}} \delta + \partial_t \delta \right), \tag{40}$$

where $\widehat{\mathrm{div}}$ denotes the Skilling–Hutchinson trace estimator introduced in Eq. (16).

## B.2. Rewriting the Conditional Distribution $\pi(\boldsymbol{a}_0 | \boldsymbol{a}_t, \boldsymbol{s})$

In this section, we derive an equivalent expression for the conditional distribution $\pi(\boldsymbol{a}_0 | \boldsymbol{a}_t, \boldsymbol{s})$. Specifically, we show that it can be written as

$$\pi(\boldsymbol{a}_0 | \boldsymbol{a}_t, \boldsymbol{s}) \propto \exp\left( \frac{1}{\alpha} Q(\boldsymbol{s}, \boldsymbol{a}_0) \right) \mathcal{N}\left( \boldsymbol{a}_0 \, \middle| \, \frac{\boldsymbol{a}_t}{1 - t}, \left( \frac{t}{1 - t} \right)^2 \boldsymbol{I} \right). \tag{41}$$

*Proof.* By Bayes' rule, we have

$$\pi(\boldsymbol{a}_0 | \boldsymbol{a}_t, \boldsymbol{s}) = \frac{\pi(\boldsymbol{a}_0 | \boldsymbol{s}) \, \pi(\boldsymbol{a}_t | \boldsymbol{a}_0)}{\pi(\boldsymbol{a}_t | \boldsymbol{s})} \tag{42}$$

$$\propto \pi(\boldsymbol{a}_0 | \boldsymbol{s}) \, \pi(\boldsymbol{a}_t | \boldsymbol{a}_0), \tag{43}$$

where $\pi(\boldsymbol{a}_t|\boldsymbol{s})$ is a constant with respect to $\boldsymbol{a}_0$.

Under the MaxEnt RL formulation, the target policy satisfies $\pi(\boldsymbol{a}_0|\boldsymbol{s}) = \frac{1}{Z(\boldsymbol{s})} \exp\left(\frac{1}{\alpha} Q(\boldsymbol{s}, \boldsymbol{a}_0)\right)$, where $Z(\boldsymbol{s})$ denotes the partition function. Substituting this expression into Eq. (43) yields

$$\pi(\boldsymbol{a}_0|\boldsymbol{a}_t, \boldsymbol{s}) \propto \exp\left(\frac{1}{\alpha} Q(\boldsymbol{s}, \boldsymbol{a}_0)\right) \pi(\boldsymbol{a}_t|\boldsymbol{a}_0). \tag{44}$$

Recall that the forward interpolation process is defined as $\boldsymbol{a}_t = (1-t)\boldsymbol{a}_0 + t\boldsymbol{a}_1$, $\boldsymbol{a}_1 \sim \mathcal{N}(\boldsymbol{0}, \boldsymbol{I})$, which implies $\pi(\boldsymbol{a}_t|\boldsymbol{a}_0) = \mathcal{N}(\boldsymbol{a}_t \mid (1-t)\boldsymbol{a}_0, t^2\boldsymbol{I})$. Note that $\mathcal{N}(\boldsymbol{a}_t \mid (1-t)\boldsymbol{a}_0, t^2\boldsymbol{I}) \propto \exp\left(-\frac{\|\boldsymbol{a}_t - (1-t)\boldsymbol{a}_0\|^2}{2t^2}\right)$, and $\mathcal{N}\left(\boldsymbol{a}_0 \mid \frac{\boldsymbol{a}_t}{1-t}, \left(\frac{t}{1-t}\right)^2 \boldsymbol{I}\right) \propto \exp\left(-\frac{\|\boldsymbol{a}_0 - \frac{\boldsymbol{a}_t}{1-t}\|^2}{2\left(\frac{t}{1-t}\right)^2}\right)$, we observe

$$\mathcal{N}(\boldsymbol{a}_t \mid (1-t)\boldsymbol{a}_0, t^2\boldsymbol{I}) \propto \mathcal{N}\left(\boldsymbol{a}_0 \mid \frac{\boldsymbol{a}_t}{1-t}, \left(\frac{t}{1-t}\right)^2 \boldsymbol{I}\right), \tag{45}$$

where the proportionality holds up to a constant independent of $\boldsymbol{a}_0$. Substituting this relation back into the conditional distribution completes the proof:

$$\pi(\boldsymbol{a}_0|\boldsymbol{a}_t, \boldsymbol{s}) \propto \exp\left(\frac{1}{\alpha} Q(\boldsymbol{s}, \boldsymbol{a}_0)\right) \mathcal{N}\left(\boldsymbol{a}_0 \mid \frac{\boldsymbol{a}_t}{1-t}, \left(\frac{t}{1-t}\right)^2 \boldsymbol{I}\right). \tag{46}$$

$\square$

### B.3. An Introduction to the Effective Sample Size of Self-Normalized Importance Sampling

Given a target distribution $p(\boldsymbol{x})$, a proposal distribution $q(\boldsymbol{x})$, and a target function $f(\boldsymbol{x})$, we aim to estimate the expectation $\mathbb{E}_p[f(\boldsymbol{x})]$. To this end, we compare the vanilla Monte Carlo (MC) estimator with the Self-Normalized Importance Sampling (SNIS) estimator:

$$\hat{\mu}_{\mathrm{MC}} = \frac{1}{K} \sum_{i=1}^{K} f(\boldsymbol{x}_i), \quad \boldsymbol{x}_i \sim p, \tag{47}$$

$$\hat{\mu}_{\mathrm{SNIS}} = \sum_{i=1}^{K} \frac{w(\boldsymbol{x}_i)}{\sum_{j=1}^{K} w(\boldsymbol{x}_j)} f(\boldsymbol{x}_i), \quad \boldsymbol{x}_i \sim q, \tag{48}$$

where $w(\boldsymbol{x}) = p(\boldsymbol{x})/q(\boldsymbol{x})$ denotes the importance weight. The efficiency of the SNIS estimator is typically quantified by the *Effective Sample Size* (ESS) (Kong, 1992; Metelli et al., 2020), which characterizes the number of i.i.d. samples from $p$ required to achieve the same estimation variance as $K$ samples from $q$. Formally, the ESS is asymptotically defined through the variance ratio:

$$\mathrm{ESS}(p \parallel q) \approx K \frac{\mathrm{Var}(\hat{\mu}_{\mathrm{MC}})}{\mathrm{Var}(\hat{\mu}_{\mathrm{SNIS}})}. \tag{49}$$

In practice, the ESS is commonly approximated using the Kish's estimator (Kong, 1992):

$$\widehat{\mathrm{ESS}}(p \parallel q) = \frac{\left(\sum_{i=1}^{K} w(\boldsymbol{x}_i)\right)^2}{\sum_{i=1}^{K} w^2(\boldsymbol{x}_i)}. \tag{50}$$

This metric ranges from 1 to $K$. It attains its maximum value $K$ when $q = p$, and degrades as the discrepancy between the proposal distribution and the target distribution increases, indicating higher estimation variance and reduced sampling efficiency.

## B.4. The Bias and Variance Analysis of the Adaptive Instantaneous Velocity Estimator

The ground-truth instantaneous velocity is given by: $v_t(a_t|s) = \mathbb{E}_{\pi(a_0|a_t,s)}\left[\frac{a_t - a_0}{t}\right]$ . We consider two self-normalized importance sampling (SNIS) estimators, $\hat{v}_t^1$ and $\hat{v}_t^2$, constructed using proposal distributions $q^1(a_0)$ and $q^2(a_0)$ with $K_1$ and $K_2$ samples, respectively. To leverage the complementary strengths of different proposals, we define a combined estimator as a linear combination of two SNIS estimators:

$$\hat{v}_t(\alpha) = \alpha\hat{v}_t^1 + (1 - \alpha)\hat{v}_t^2, \qquad \alpha \in [0, 1]. \tag{51}$$

**Asymptotic Bias.** It is a well-known property that SNIS estimators are biased in finite samples but asymptotically unbiased. Specifically, for each estimator $k \in \{1, 2\}$, its expectation satisfies (Owen, 2013):

$$\mathbb{E}[\hat{v}_t^k] = v_t + \mathcal{O}(K_k^{-1}). \tag{52}$$

Consequently, as a linear combination with weights summing to one, the combined estimator $\hat{v}_t(\alpha)$ remains asymptotically unbiased, with the bias vanishing at a rate of $\mathcal{O}(\min(K_1, K_2)^{-1})$.

**Variance.** Assuming the two SNIS estimators are independent, the variance of the combined estimator is:

$$\mathrm{Var}[\hat{v}_t(\alpha)] = \alpha^2\mathrm{Var}[\hat{v}_t^1] + (1 - \alpha)^2\mathrm{Var}[\hat{v}_t^2]. \tag{53}$$

Minimizing Eq. (53) with respect to $\alpha$ yields the optimal weight:

$$\alpha^\star = \frac{\mathrm{Var}[\hat{v}_t^2]}{\mathrm{Var}[\hat{v}_t^1] + \mathrm{Var}[\hat{v}_t^2]}. \tag{54}$$

Thus, the minimum-variance estimator should assign larger weights to the component with smaller variance. For SNIS, the variance is approximately inversely proportional to the ESS:

$$\mathrm{Var}[\hat{v}_t^k] \propto \frac{1}{\mathrm{ESS}_k}. \tag{55}$$

Substituting this approximation into Eq. (54), we derive the adaptive ESS-weighted estimator employed in MFPO:

$$\hat{v}_t = \frac{\mathrm{ESS}_1}{\mathrm{ESS}_1 + \mathrm{ESS}_2}\hat{v}_t^1 + \frac{\mathrm{ESS}_2}{\mathrm{ESS}_1 + \mathrm{ESS}_2}\hat{v}_t^2. \tag{56}$$

This ESS-weighted estimator is asymptotically unbiased and achieves near-optimal variance among linear combinations of the two SNIS estimators. As a result, the estimator automatically favors the proposal that better matches the target distribution, providing a stable and efficient instantaneous velocity estimation for MFPO.

## B.5. Convergence Analysis of MFPO

**Soft Policy Evaluation.** We first show that given an accurate likelihood estimator, minimizing the soft Bellman error recovers the soft Q-function of the current policy.

**Proposition B.1** (Soft Policy Evaluation). *Assume that the Average Divergence Network (ADN) loss in Eq. (17) is minimized such that the resulting ADN recovers the true action likelihood $\log \pi_\theta(a|s)$ of the current MeanFlow policy. Further assume that the soft Bellman error in Eq. (29) is minimized over a sufficiently expressive function class. Then the learned Q-function $Q_\phi$ converges to the true soft Q-function $Q^{\pi_\theta}$ of the current policy $\pi_\theta$.*

*Proof.* For a MeanFlow policy, the log-likelihood of an action can be computed through the instantaneous change-of-variables formula, which depends on the divergence of the instantaneous velocity field along the transport trajectory. The Skilling–Hutchinson trace estimator provides an unbiased estimator of this divergence. Therefore, when the ADN loss

$$\mathcal{L}(\omega) = \mathbb{E}_{s,r,t,a_t,\epsilon_{1:N}} \|\delta_\omega(s, a_t, r, t) - \mathrm{sg}(\delta_{\mathrm{tgt}})\|_2^2$$

is minimized, the learned ADN $\delta_\omega$ recovers the corresponding average divergence. Consequently, the resulting likelihood estimate $\log \pi_\theta(a \mid s)$ is accurate.

Given the accurate likelihood of the current policy, the soft Bellman target is

$$Q_{\text{target}}(\boldsymbol{s}, \boldsymbol{a}) = r(\boldsymbol{s}, \boldsymbol{a}) + \gamma \mathbb{E}_{\boldsymbol{s}' \sim p(\cdot | \boldsymbol{s}, \boldsymbol{a}), \boldsymbol{a}' \sim \pi_\theta(\cdot | \boldsymbol{s}')} \left[ Q_\phi(\boldsymbol{s}', \boldsymbol{a}') - \alpha \log \pi_\theta(\boldsymbol{a}' | \boldsymbol{s}') \right].$$

Thus, minimizing the soft Bellman error

$$\mathcal{L}(\phi) = \mathbb{E}_{(\boldsymbol{s}, \boldsymbol{a}, \boldsymbol{s}') \sim \mathcal{B}} \left[ \frac{1}{2} \left( Q_\phi(\boldsymbol{s}, \boldsymbol{a}) - Q_{\text{target}}(\boldsymbol{s}, \boldsymbol{a}) \right)^2 \right]$$

enforces $Q_\phi$ to be a fixed point of the soft Bellman operator associated with $\pi_\theta$. Since this operator is a contraction (Haarnoja et al., 2017), it admits a unique fixed point, which is precisely the soft Q-function $Q^{\pi_\theta}$. Therefore, at the global minimum, $Q_\phi = Q^{\pi_\theta}$. $\qquad\square$

**Soft Policy Improvement.** We next show that the policy improvement step recovers the Boltzmann policy induced by the current soft Q-function when the Adaptive Instantaneous Velocity Estimation (AIVE) is unbiased.

**Proposition B.2** (Soft Policy Improvement). *Let $K_1$ and $K_2$ denote the sample sizes of the two SNIS estimators used in AIVE. Suppose that $K_1, K_2 \to \infty$, such that AIVE provides an unbiased estimator of the instantaneous velocity field associated with the target distribution*

$$\pi^*(\boldsymbol{a} | \boldsymbol{s}) = \frac{\exp\left(\frac{1}{\alpha} Q^{\pi_{\text{old}}}(\boldsymbol{s}, \boldsymbol{a})\right)}{Z(\boldsymbol{s})}.$$

*Assume further that the MeanFlow policy class $\Pi$ is sufficiently expressive such that $\pi^*(\cdot | \boldsymbol{s}) \in \Pi$. Then minimizing the MeanFlow policy loss in Eq. (28) yields $\pi_{\text{new}} = \pi^*$, which improves upon $\pi_{\text{old}}$ in the MaxEnt objective.*

*Proof.* From the bias analysis of AIVE in App. B.4, as $K_1, K_2 \to \infty$, AIVE provides an unbiased estimator of the instantaneous velocity field corresponding to the transport toward the target policy $\pi^*(\cdot | \boldsymbol{s})$. Minimizing the MeanFlow policy loss

$$\mathcal{L}(\theta) = \mathbb{E}_{\boldsymbol{s}, r, t, \boldsymbol{a}_t} \| \boldsymbol{u}_\theta(\boldsymbol{s}, \boldsymbol{a}_t, r, t) - \text{sg}(\boldsymbol{u}_{\text{tgt}}) \|_2^2$$

therefore recovers the average velocity field whose induced terminal distribution is $\pi^*(\cdot | \boldsymbol{s})$. Consequently, the updated MeanFlow policy satisfies

$$\pi_{\text{new}}(\cdot | \boldsymbol{s}) = \pi^*(\cdot | \boldsymbol{s}).$$

By the standard soft policy improvement theorem, the Boltzmann policy $\pi^*$ induced by $Q^{\pi_{\text{old}}}$ minimizes the KL objective in Eq. (4) and therefore satisfies

$$Q^{\pi_{\text{new}}}(\boldsymbol{s}, \boldsymbol{a}) \geq Q^{\pi_{\text{old}}}(\boldsymbol{s}, \boldsymbol{a}), \quad \forall (\boldsymbol{s}, \boldsymbol{a}),$$

under the MaxEnt objective. Hence the policy improvement step improves upon $\pi_{\text{old}}$. $\qquad\square$

**Soft Policy Iteration.** According to the standard soft policy iteration theorem in Sec. 3.2, by repeatedly alternating between soft policy evaluation and soft policy improvement, MFPO will converge to the optimal MaxEnt policy within $\Pi$.

# C. Algorithm Pseudo Code

We present the training procedure for the average divergence network, the likelihood-aware sampling procedure, and the complete MFPO training algorithm in Alg. 1, 2, and 3, respectively.

# D. Experimental Details

In this paper, all experiments are conducted on a server equipped with an NVIDIA GeForce RTX 3090 GPU and an Intel Xeon Platinum 8280 CPU. The implementations of baseline methods follow their official repositories: DIME[1], FlowRL[2],

---

[1] https://github.com/ALRhub/DIME
[2] https://github.com/bytedance/FlowRL

---

**Algorithm 1** Training of the Average Divergence Network

---

1: **Input:** policy network $\boldsymbol{u}_\theta$; average divergence network $\delta_\omega$; a batch of $B$ state-action pairs $\{(\boldsymbol{s}, \boldsymbol{a}_0)\}$ from the current policy
2: Sample $(r, t) \sim p(r, t)$ and $\boldsymbol{a}_1 \sim \mathcal{N}(\boldsymbol{0}, \boldsymbol{I})$
3: $\boldsymbol{a}_t \leftarrow (1 - t)\boldsymbol{a}_0 + t\boldsymbol{a}_1$
4: **for** each $\boldsymbol{a}_t$ in the batch **do**
5: $\quad$ Sample $\{\boldsymbol{\epsilon}_i\}_{i=1}^N \sim \mathcal{N}(\boldsymbol{0}, \boldsymbol{I})$
6: $\quad$ Estimate divergence: $\widehat{\text{div}}(\boldsymbol{s}, \boldsymbol{a}_t, t) = \frac{1}{N} \sum_{i=1}^N \boldsymbol{\epsilon}_i^\top \frac{\partial \boldsymbol{u}_\theta(\boldsymbol{s}, \boldsymbol{a}_t, t, t)}{\partial \boldsymbol{a}_t} \boldsymbol{\epsilon}_i$
7: $\quad$ Construct target: $\delta_{\text{tgt}}(\boldsymbol{s}, \boldsymbol{a}_t, r, t) = \widehat{\text{div}} - (t - r)\left(\boldsymbol{u}_\theta(\boldsymbol{s}, \boldsymbol{a}_t, t, t)\,\partial_{\boldsymbol{a}}\delta + \partial_t\delta\right)$
8: **end for**
9: Update parameters: $\omega \leftarrow \arg\min_\omega \frac{1}{B} \sum \|\delta_\omega(\boldsymbol{s}, \boldsymbol{a}_t, r, t) - \text{sg}(\delta_{\text{tgt}}(\boldsymbol{s}, \boldsymbol{a}_t, r, t))\|_2^2$

---

**Algorithm 2** MeanFlow Action Sampling and Likelihood Estimation

---

1: **Input:** state $\boldsymbol{s}$; policy network $\boldsymbol{u}_\theta$; average divergence network $\delta_\omega$; number of sampling steps $T$; prior distribution $p_1$
2: Sample $\boldsymbol{a}_{t_T} \sim p_1(\boldsymbol{a}_1)$
3: Initialize $\log p(\boldsymbol{a}_{t_T}) \leftarrow \log p_1(\boldsymbol{a}_1)$
4: **for** $i = T, \ldots, 1$ **do**
5: $\quad \boldsymbol{a}_{t_{i-1}} \leftarrow \boldsymbol{a}_{t_i} - \frac{1}{T} \boldsymbol{u}_\theta(\boldsymbol{s}, \boldsymbol{a}_{t_i}, t_{i-1}, t_i)$
6: $\quad \log p(\boldsymbol{a}_{t_{i-1}}) \leftarrow \log p(\boldsymbol{a}_{t_i}) + \frac{1}{T} \delta_\omega(\boldsymbol{s}, \boldsymbol{a}_{t_i}, t_{i-1}, t_i)$
7: **end for**
8: **Output:** action $\boldsymbol{a}_0$ and log-likelihood $\log p(\boldsymbol{a}_0)$

---

SAC-Flow[3], MaxEntDP[4], DACER[5], QVPO[6], TD3[7], and SAC[8]. These official implementations are developed using different machine learning frameworks. Specifically, DIME, SAC-Flow, MaxEntDP, and DACER are implemented in JAX (Frostig et al., 2019), FlowRL is implemented in PyTorch (Paszke et al., 2019) and executed in compiled mode with performance comparable to JAX, while QVPO, SAC, and TD3 are implemented in PyTorch. To ensure a fair comparison of training and inference efficiency, the reported training and inference time of QVPO, SAC, and TD3 is measured using our JAX implementations of these methods. The hyperparameters shared across all algorithms are summarized in Tab. 2. All hyperparameters follow their default settings in the official implementations, except that the batch size, the network size and the update-to-data ratio are unified across all methods for fair comparison.

*Table 2.* Shared hyperparameters across all algorithms.

| Hyperparameter | MFPO | DIME | FlowRL | SAC-Flow | MaxEntDP | DACER | QVPO | TD3 | SAC |
|---|---|---|---|---|---|---|---|---|---|
| Batch size | 256 | 256 | 256 | 256 | 256 | 256 | 256 | 256 | 256 |
| Discount factor $\gamma$ | 0.99 | 0.99 | 0.99 | 0.99 | 0.99 | 0.99 | 0.99 | 0.99 | 0.99 |
| Target smoothing coefficient $\tau$ | 0.005 | 1 | 0.95 | 1 | 0.005 | 0.005 | 0.005 | 0.005 | 0.005 |
| No. of hidden layers | 3 | 3 | 3 | N/A | 3 | 3 | 3 | 3 | 3 |
| No. of hidden nodes per layer | 256 | 256 | 256 | N/A | 256 | 256 | 256 | 256 | 256 |
| Actor learning rate | 3e-4 | 3e-4 | 3e-4 | 3e-4 | 3e-4 | 1e-4 | 3e-4 | 3e-4 | 3e-4 |
| Critic learning rate | 3e-4 | 3e-4 | 3e-4 | 1e-3 | 3e-4 | 1e-4 | 3e-4 | 3e-4 | 3e-4 |
| Activation function | gelu | gelu | elu | gelu | mish | mish | mish | relu | relu |
| Replay buffer size | 1e6 | 1e6 | 1e6 | 1e6 | 1e6 | 1e6 | 1e6 | 1e6 | 1e6 |
| Update-to-data ratio | 1 | 1 | 1 | 1 | 1 | 1 | 1 | 1 | 1 |
| Diffusion steps | 2 | 16 | 1 | 4 | 20 | 20 | 20 | N/A | N/A |
| No. of action candidates | 10 | N/A | N/A | N/A | 10 | N/A | 32 | N/A | N/A |

---

[3] https://github.com/Elessar123/SAC-FLOW
[4] https://github.com/diffusionyes/MaxEntDP
[5] https://github.com/happy-yan/DACER-Diffusion-with-Online-RL
[6] https://github.com/wadx2019/qvpo
[7] https://github.com/sfujim/TD3
[8] https://github.com/toshikwa/soft-actor-critic.pytorch

---

**Algorithm 3** Mean Flow Policy Optimization (MFPO)

1: **Input:** Q-network $Q_\phi$; target Q-network $Q_{\bar\phi}$; policy network $\boldsymbol{u}_\theta$; average divergence network $\delta_\omega$; temperature $\alpha$
2: **for** each iteration **do**
3:     **for** each interaction step **do**
4:         Sample $\boldsymbol{a} \sim \pi_\theta(\cdot|\boldsymbol{s})$ using Algorithm 2
5:         Execute action and observe $(\boldsymbol{s}', r) = \text{env}(\boldsymbol{a})$
6:         Store transition $(\boldsymbol{s}, \boldsymbol{a}, r, \boldsymbol{s}')$ in replay buffer $\mathcal{B}$
7:     **end for**
8:     **for** each update step **do**
9:         Sample $B$ transitions $\{(\boldsymbol{s}, \boldsymbol{a}, r, \boldsymbol{s}')\}$ from $\mathcal{B}$
10:         *// Policy evaluation:*
11:         Sample $\boldsymbol{a}' \sim \pi_\theta(\cdot|\boldsymbol{s}')$ and compute $\log \pi_\theta(\boldsymbol{a}'|\boldsymbol{s}')$ using Algorithm 2
12:         Compute target: $Q_{\text{target}} \leftarrow r + \gamma \left( Q_{\bar\phi}(\boldsymbol{s}', \boldsymbol{a}') - \alpha \log \pi_\theta(\boldsymbol{a}'|\boldsymbol{s}') \right)$
13:         Update Q-network: $\phi \leftarrow \arg\min_\phi \frac{1}{B} \sum \frac{1}{2} \left( Q_\phi(\boldsymbol{s}, \boldsymbol{a}) - Q_{\text{target}} \right)^2$
14:         *// Policy improvement:*
15:         Sample $(r, t) \sim p(r, t)$
16:         **for** $k = 1, 2$ **do**
17:             Sample $\boldsymbol{a}_0 \sim q^k$
18:             Estimate instantaneous velocity via SNIS: $\hat{\boldsymbol{v}}_t^k = \sum \frac{w(\boldsymbol{a}_0)}{\sum w(\boldsymbol{a}_0)} \cdot \frac{\boldsymbol{a}_t - \boldsymbol{a}_0}{t}$
19:             Compute effective sample size: $\text{ESS}_k = \frac{(\sum w(\boldsymbol{a}_0))^2}{\sum w^2(\boldsymbol{a}_0)}$
20:         **end for**
21:         Combine estimates: $\hat{\boldsymbol{v}}_t = \frac{\text{ESS}_1}{\text{ESS}_1 + \text{ESS}_2} \hat{\boldsymbol{v}}_t^1 + \frac{\text{ESS}_2}{\text{ESS}_1 + \text{ESS}_2} \hat{\boldsymbol{v}}_t^2$
22:         Construct target: $\boldsymbol{u}_{\text{tgt}} = \hat{\boldsymbol{v}}_t - (t - r)\left(\hat{\boldsymbol{v}}_t \partial_{\boldsymbol{a}} \boldsymbol{u}_\theta + \partial_t \boldsymbol{u}_\theta\right)$
23:         Update policy network: $\theta \leftarrow \arg\min_\theta \frac{1}{B} \sum \|\boldsymbol{u}_\theta(\boldsymbol{s}, \boldsymbol{a}_t, r, t) - \text{sg}(\boldsymbol{u}_{\text{tgt}})\|_2^2$
24:         Update the average divergence network $\delta_\omega$ using Algorithm 1
25:         Update target network: $\bar\phi \leftarrow \tau\phi + (1 - \tau)\bar\phi$
26:     **end for**
27: **end for**

---

# E. Supplementary Experiments

### E.1. 2-D Toy Example

We conduct experiments on a 2-D toy example to validate the effectiveness of the proposed MFPO method. Specifically, the policy network and the average divergence network are trained to approximate a Gaussian Mixture Model (GMM) with 6 symmetrically placed components, where the reward function is defined as the log-probability of the target GMM. After training, we visualize the actions generated by the MeanFlow policy together with their log-likelihood estimates obtained from the average divergence network, as shown in Fig. 5(c). For reference, Fig. 5(a) displays actions sampled from the true GMM along with their exact log-likelihoods, while Fig. 5(b) presents the actions and log-likelihood estimates obtained by solving the diffusion ODE using a 1000-step Euler solver. As observed, the action distribution and log-likelihood estimates produced by the 2-step MFPO closely match those of the true GMM and the fine-grained Euler solver. This result demonstrates that MFPO can accurately approximate both the target action distribution and its likelihood using only a few sampling steps, highlighting its effectiveness.

### E.2. Computation Time of Key Components in MFPO

We evaluate the computational overhead and performance gains of the key components in MFPO, including MeanFlow models, the Average Divergence Network (ADN), and Adaptive Instantaneous Velocity Estimation (AIVE). We begin with a diffusion-policy baseline obtained by removing all three components, where the algorithm learns a standard Q-function and trains an instantaneous velocity network to approximate the SNIS estimator under a single Gaussian proposal. We then progressively incorporate each component to assess its contribution in terms of both computational cost and policy performance. The returns of all variants are normalized by the return of the full MFPO algorithm. The results are reported in Tab. 3. Overall, the proposed components incur only a little computational overhead while providing substantial performance

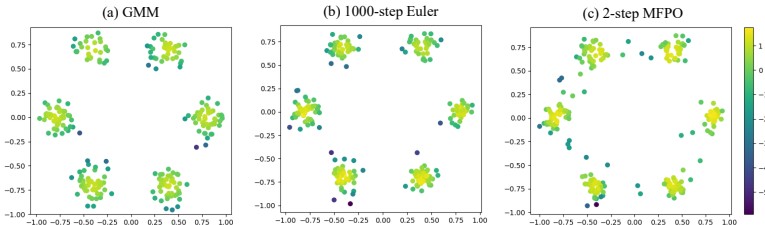

*Figure 5.* Visualization of generated actions and their log-likelihood estimates on a 2-D toy example. Each point denotes an action, and the color indicates the corresponding log-likelihood value. (a) Actions and exact log-likelihoods from the true GMM. (b) Actions and log-likelihood estimates obtained by solving the diffusion ODE with a 1000-step Euler solver. (c) Actions and log-likelihood estimates produced by the 2-step MFPO.

improvements.

*Table 3.* Training overhead and performance gain of key components in MFPO, evaluated on Mujoco benchmarks.

| Algorithm | Training Time (h) | Normalized Return |
|---|---|---|
| Diffusion Policy | 2.45 | 0.52 |
| MeanFlow Policy | 2.45 | 0.77 |
| MeanFlow Policy + ADN | 2.58 | 0.91 |
| MeanFlow Policy + ADN + AIVE | 2.81 | 1.00 |

### E.3. The approximation error of the Average Divergence Network

Effective soft policy evaluation critically relies on accurate action-likelihood estimation by the Average Divergence Network (ADN). In Fig. 6, we report the ADN training loss, the variance of the Skilling–Hutchinson trace estimator, and the relative error of the action-likelihood estimates produced by ADN against those obtained from a 1000-step Euler ODE solver, which serves as a near-ground-truth reference. The results show that both the training loss and estimator variance remain low throughout training, indicating stable optimization. Moreover, ADN closely matches the likelihood estimates of the 1000-step Euler solver while using only two sampling steps, enabling efficient and accurate likelihood estimation for soft policy evaluation.

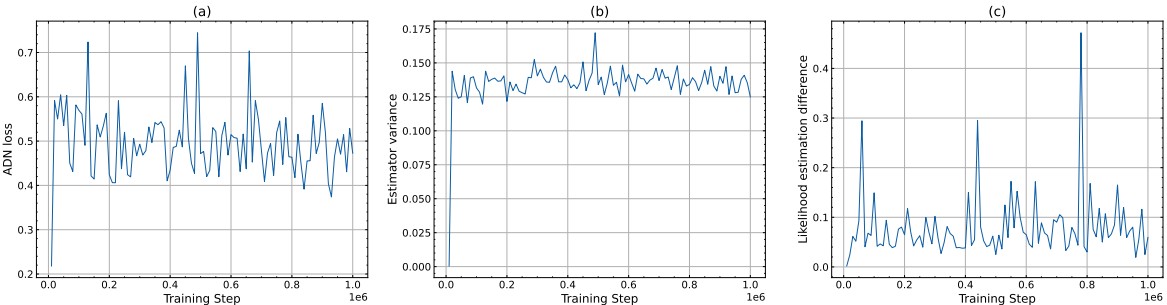

*Figure 6.* Training analysis of the Average Divergence Network (ADN) on the HalfCheetah-v3 benchmark. (a) Training loss of ADN. (b) Estimation variance of the Skilling–Hutchinson trace estimator. (c) Relative error of action likelihood estimation compared to a 1000-step Euler ODE solver, normalized by the latter.

### E.4. Comparative Evaluation on DeepMind Control Suite and HumanoidBench

We compare MFPO with baseline algorithms on 6 hard tasks from DeepMind Control Suite and 3 high-dimensional control tasks from HumanoidBench. As shown in Fig. 7 and Fig. 8, MFPO consistently achieves comparable or better performance than diffusion-based baselines, while requiring less training time.

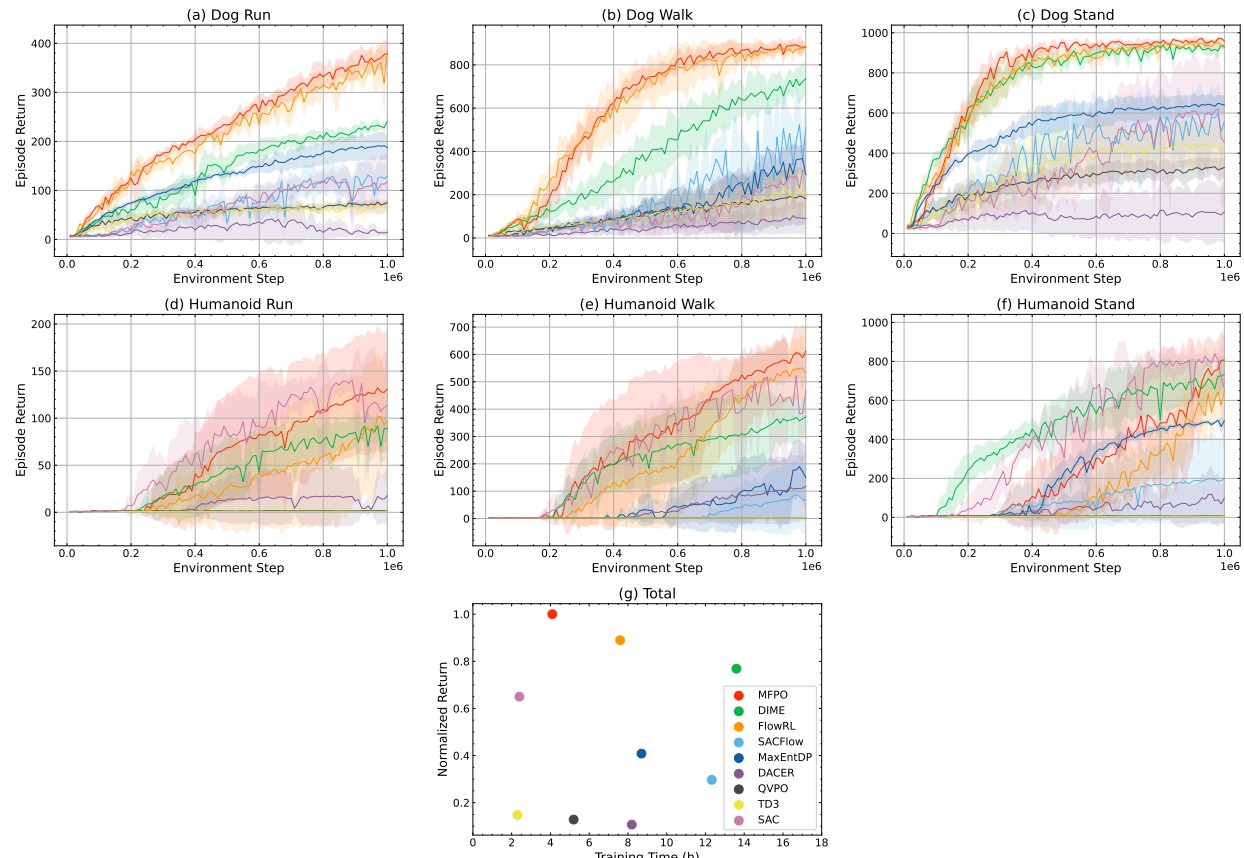

*Figure 7.* Comparison of different algorithms on 6 hard tasks from DeepMind Control Suite. (a–f) Learning curves averaged over 5 random seeds; the shaded regions denote the standard deviation. (g) Average normalized return versus training time across all tasks. The normalized return on each task is computed by dividing the average return over the last 10% of environment steps by that of MFPO. Methods closer to the upper-left region exhibit both higher performance and greater time efficiency.

### E.5. Comparison with Gaussian Mixture Models and Normalization Flows

In addition to the MeanFlow models adopted in this paper, other generative models can also represent multi-modal distributions, such as Gaussian mixture models and normalizing flows. We compare MFPO with GMM-based policies and normalizing-flow-based policies to demonstrate the advantages of MeanFlow models as policy representations. Specifically, we implement SAC-GMM, a variant of SAC whose policy is parameterized as a Gaussian mixture, and evaluate it with 4 and 16 mixture components. We also compare against MEow (Chao et al., 2024), a MaxEnt RL algorithm that represents policies using normalizing flows.

As shown in Figure 9, MFPO outperforms both SAC-GMM and MEow. For GMM-based policies, the appropriate number of mixture components is task-dependent and difficult to determine in practice. For normalizing flows, the requirement of invertible transformations constrains the class of network parameterization, which may limit their expressivity. These limitations may partly explain the observed performance gap and highlight the effectiveness of MeanFlow models as expressive policy representations for MaxEnt RL.

### E.6. Multi-modal Policy Learning on AntMaze Benchmarks

In this subsection, we investigate whether MFPO can learn multi-modal policies on complex multi-goal RL tasks. To highlight the importance of combining the MaxEnt RL objective with an expressive policy representation, we compare MFPO against SAC and BPTT. Here, BPTT is a simple baseline implemented by ourselves, which trains a MeanFlow policy under the standard RL objective by directly backpropagating Q-gradients through the flow chain. We evaluate these algorithms on AntMaze benchmarks, where algorithms require to control a quadruped robot to navigate from the start to any goal.

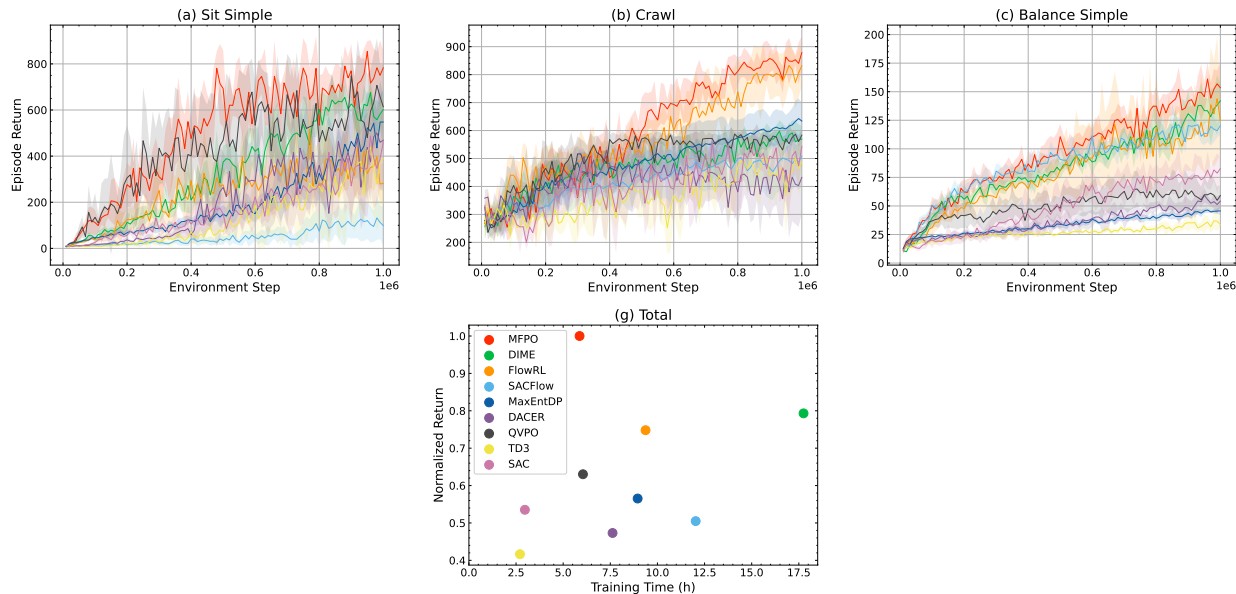

*Figure 8.* Comparison of different algorithms on 3 high-dimensional tasks from HumanoidBench. (a–f) Learning curves averaged over 5 random seeds; the shaded regions denote the standard deviation. (g) Average normalized return versus training time across all tasks. The normalized return on each task is computed by dividing the average return over the last 10% of environment steps by that of MFPO. Methods closer to the upper-left region exhibit both higher performance and greater time efficiency.

Figure 10 visualizes the trajectories generated by different algorithms. MFPO discovers multiple feasible paths, demonstrating its ability to learn multi-modal policies in challenging multi-goal tasks. Although SAC optimizes the MaxEnt RL objective to encourage diverse behaviors, the unimodal nature of Gaussian policies limits it to a single behavior mode. As a result, SAC may converge to only one feasible solution or even fail to discover a valid solution, as observed on AntMaze-v4. These results suggest that expressive policy representations capable of modeling complex multi-modal action distributions are crucial for exploring multiple behavior modes effectively.

However, expressivity alone is not sufficient. Without entropy regularization, BPTT tends to collapse to near-deterministic solutions due to pure Q maximization. Indeed, under the standard RL objective, the optimal policy places all probability mass on actions that maximize $Q(s, a)$, rather than maintaining diverse action modes. Overall, the combination of MeanFlow policies and the MaxEnt RL framework is necessary: MeanFlow model provides the expressive policy class required to represent multi-modal behaviors, while the MaxEnt objective prevents mode collapse and encourages the discovery of diverse feasible solutions.

### E.7. Ablation Studies

**Distributional Q-learning.** Since the policy improvement step is guided by the Q function obtained during policy evaluation, accurately estimating the Q function of the current policy is critical for effective policy optimization. As shown in Fig. 11(a), disabling distributional Q-learning leads to a performance degradation of MFPO. This result indicates that enhancing the accuracy of policy evaluation via distributional Q-learning plays a significant role in the overall performance of MFPO.

**Action Selection.** At inference time, the action selection technique can further improve the optimality of actions generated by MeanFlow policies. We evaluate the effect of different numbers of action candidates $M$ in Fig. 11(b). The results show that employing action selection ($M > 1$) outperforms the setting without action selection ($M = 1$). Therefore, we set $M = 10$ in all experiments, as it achieves strong performance while incurring only a modest computational overhead.

### E.8. Hyperparameter Analysis

Here we analyze the effect of key hyperparameters on the performance of MFPO.

**Sampling steps.** By modeling the average velocity field, MeanFlow policies retain high expressivity even with a small number of sampling steps. As shown in Fig. 12(a), using only two sampling steps achieves performance comparable to that

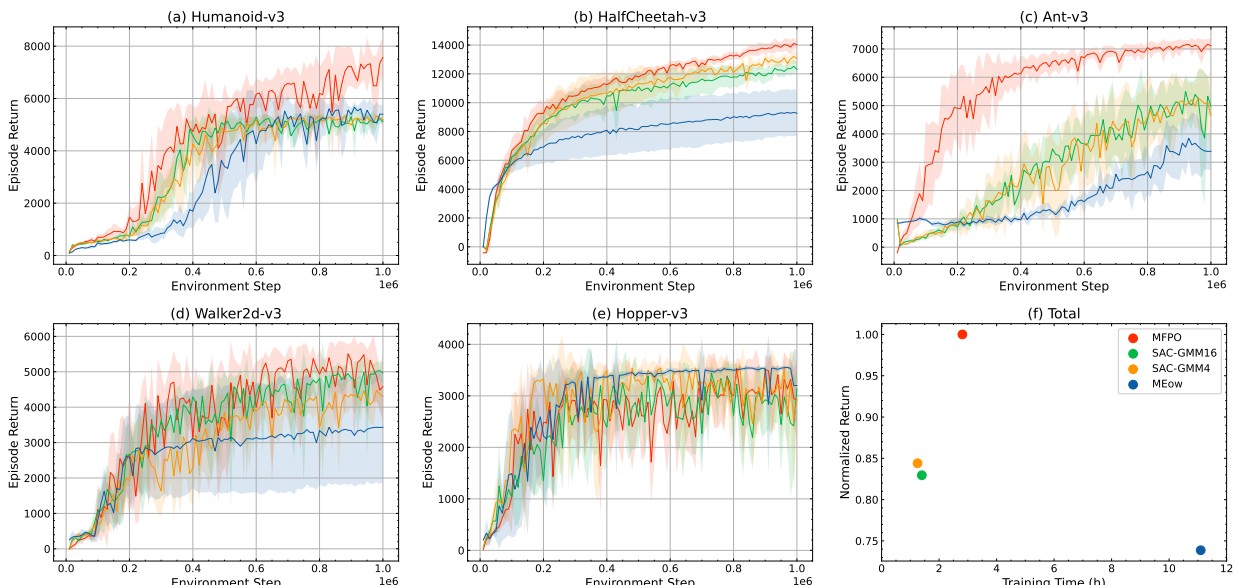

*Figure 9.* Comparison with SAC-GMM and MEow on MuJoCo benchmarks. (a–f) Learning curves averaged over 5 random seeds; the shaded regions denote the standard deviation. (g) Average normalized return versus training time across all tasks. The normalized return on each task is computed by dividing the average return over the last 10% of environment steps by that of MFPO. Methods closer to the upper-left region exhibit both higher performance and greater time efficiency.

obtained with a much larger number of steps (e.g., $T = 16$), demonstrating the advantage of adopting MeanFlow models as policy representations. Throughout this paper, we therefore set $T = 2$ as the default configuration.

**Sample number for instantaneous velocity estimation.** Increasing the number of samples reduces both the bias and variance of the SNIS estimator, and is thus preferable when computational resources permit. This trend is consistent with the empirical results in Fig. 12(b), where larger sample sizes lead to improved performance. Based on this trade-off, we set $K_1 = 16$ and $K_2 = 32$, which perform well across most tasks.

**Sample number for divergence estimation.** The variance of the divergence estimator decreases as the number of samples $N$ increases. Consequently, using more samples yields more stable divergence estimates, which in turn stabilizes the training of the average divergence network and improves action likelihood estimation. As shown in Fig. 12(c), setting $N = 2$ already achieves strong performance, while further increasing $N$ results in only marginal improvements.

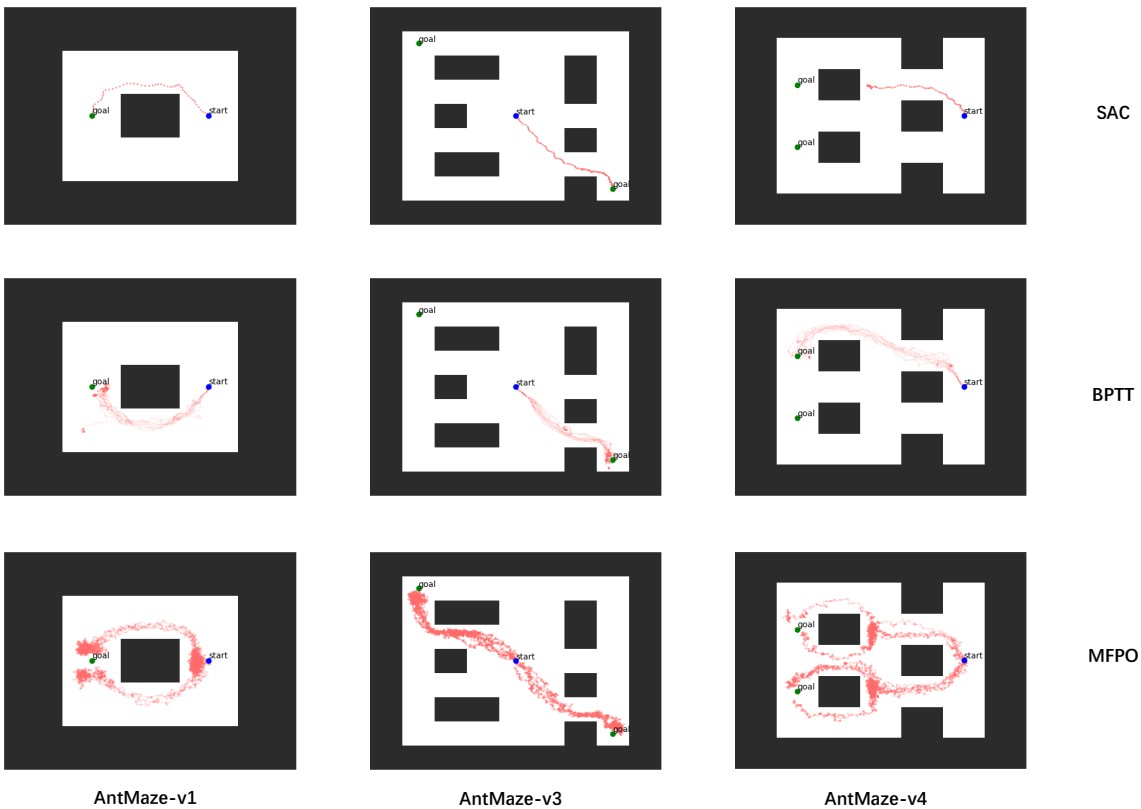

*Figure 10.* Trajectories generated by SAC, BPTT, and MFPO after 100k environment interactions on AntMaze tasks.

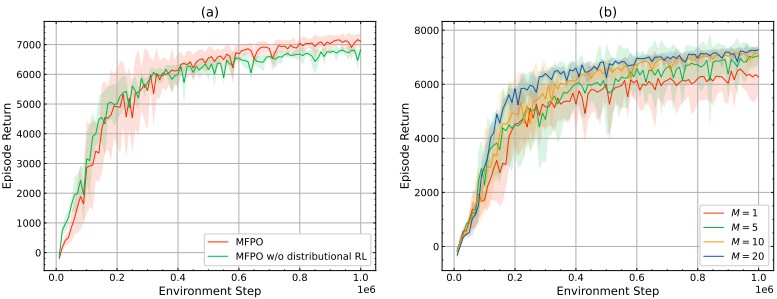

*Figure 11.* Ablation study of distributional Q-learning and action selection on the Ant-v3 benchmark. (a) Learning curves of MFPO with and without distributional Q-learning. (b) Learning curves of MFPO with different numbers of candidate actions.

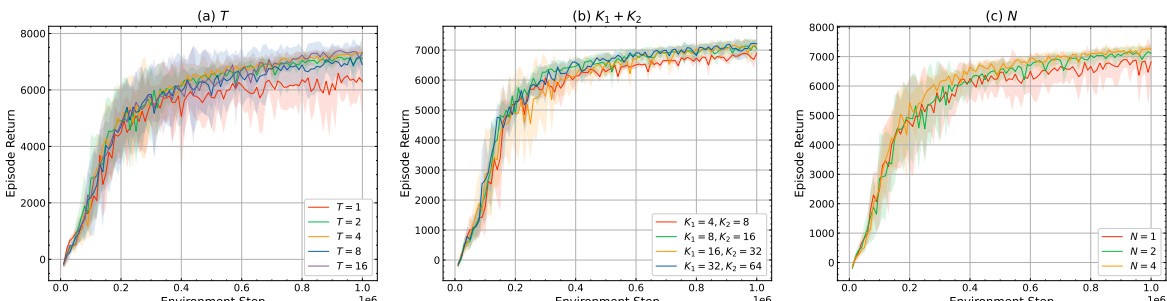

*Figure 12.* Hyperparameter analysis on the Ant-v3 benchmark. (a) Learning curves of MFPO with different numbers of sampling steps $T$. (b) Learning curves of MFPO with different numbers of samples for instantaneous velocity estimation; the sample ratio between the two proposals is fixed while their total number is varied. (c) Learning curves of MFPO with different numbers of samples $N$ for divergence estimation.

