# OpenReview forum: "Mean Flow Policy Optimization"
_ICML.cc/2026/Conference — ICML 2026 regular_

### Official Review · Reviewer_iTvu · 2026-03-03

**Soundness:** 4
**Presentation:** 4
**Significance:** 2
**Originality:** 3
**Overall Recommendation:** 4
**Confidence:** 4

**Summary:**

This paper proposes Mean Flow Policy Optimization (MFPO), a method that integrates MeanFlow models into the maximum entropy (MaxEnt) RL framework for continuous control. The authors adopt MeanFlow models to enable high-quality action generation with very few sampling steps. Furthermore, to tackle the intractable action likelihood evaluation in diffusion-based policies, the authors introduce an Average Divergence Network (ADN). The proposed algorithm is evaluated on standard benchmarks, demonstrating performance comparable to existing diffusion-based RL methods but with significantly reduced training and inference times.

**Compliance With Llm Reviewing Policy:**

Affirmed.

**Key Questions For Authors:**

1. Can the authors provide evaluations on more complex, modern, or inherently multi-modal experimental environments？
2. To alleviate concerns regarding the stability of the joint alternating training, could the authors provide empirical evidence, such as learning curves tracking the ADN loss or the variance of the Skilling-Hutchinson trace estimator throughout the RL training process?

**Limitations:**

yes

**Strengths And Weaknesses:**

### Strengths
1. Novel Integration and Effective Solution to a Known Bottleneck: The paper successfully introduces the MeanFlow matching framework into the RL domain. More importantly, the proposed Average Divergence Network (ADN) effectively and elegantly solves a notorious pain point of applying flow matching/diffusion to RL: the inability to explicitly and efficiently compute the exact probability distribution (action likelihood) of generated actions.

2. Solid Theoretical Foundation: The theoretical derivations—from the MeanFlow identity to the formulation of the ADN training objective and the self-normalized importance sampling (SNIS) for velocity estimation—are rigorously grounded and mathematically sound.

3. Clear Presentation and Thorough Ablations: The paper is exceptionally well-written, clear, and easy to follow. The ablation studies are comprehensive, well-designed, and successfully justify the core components of the proposed method (e.g., the necessity of ADN, the choice of sampling ratios, and temperature tuning).

### Weaknesses
1. Limited Empirical Evaluation and Marginal Gains: The chosen experimental environments (standard MuJoCo and DeepMind Control Suite locomotion tasks) are relatively simple and somewhat outdated for evaluating the expressivity of state-of-the-art generative RL policies. Furthermore, despite the introduction of a sophisticated generative algorithm and the overhead of training an additional network (ADN), the final performance improvements over existing diffusion-based baselines—and even strong traditional baselines like SAC or TD3—appear marginal.

2. Concerns Regarding Training Stability and Convergence: The joint, alternating training of the policy network and the ADN seems potentially unstable. Because the ADN's training target is continuously shifting based on the evolving policy's velocity field, I am highly skeptical about whether the ADN can consistently and effectively converge, especially during the early phases of RL training when the policy is changing rapidly.

3. Questionable significance of the claimed challenge: The authors motivate the use of MeanFlow to address specific challenges in RL. However, I suspect the actual significance of the challenge that MeanFlow aims to solve within the standard RL locomotion benchmarks evaluated in the paper.

---

> ### Author Rebuttal · Authors · 2026-03-31
>
> **W1.1: Limited empirical evaluation.**
>
> **A1.1**: We further evaluate MFPO and competing methods on HumanoidBench [1] (with a 61-dimensional action space), with results shown in Fig. 15 of https://anonymous.4open.science/r/MFPO-pics/HumanoidBench.pdf. On HumanoidBench, MFPO consistently outperforms SAC, FlowRL, and DIME across all tasks, demonstrating its effectiveness in high-dimensional control tasks.
>
> In addition, we compare MFPO with SAC and BPTT on AntMaze [2] benchmarks (multi-modal policy learning tasks), with results shown in Fig. 14 of https://anonymous.4open.science/r/MFPO-pics/BPTT.pdf. BPTT is a naive baseline developed ourselves that optimizes MeanFlow policies under the standard RL objective. Specifically, BPTT learns the standard Q-function and updates the policy by backpropagating Q-gradients through the flow rollout. On AntMaze tasks, MFPO successfully explores multiple feasible paths, whereas BPTT and SAC are limited to a single mode.
>
> **W1.2: Marginal gains**
>
> **A1.2:** Our goal is to reduce the training and inference cost of diffusion/flow-based policies without sacrificing performance. As shown in Fig. 2, Fig. 6, and Tab. 1, MFPO achieves ~2× faster training and ~3–4× faster inference compared to diffusion-based baselines, while maintaining comparable or superior performance.
>
> **W2: Training stability and convergence.**
>
> **A2:** In Fig. 11 of https://anonymous.4open.science/r/MFPO-pics/ADN.pdf, we report the training loss of ADN, the variance of the Skilling–Hutchinson trace estimator, and the relative error of action likelihood estimation between ADN and a 1000-step Euler ODE solver (used as a near ground-truth reference). The results show that the training loss and the estimator variance remain low throughout training, indicating stable optimization behavior.
>
> In addition, ADN closely matches the likelihood estimates of the 1000-step Euler solver while requiring only two steps, enabling efficient and accurate likelihood estimation for soft policy evaluation.
>
> **W3: The significance of the claimed challenge.**
>
> **A3:** We elaborate on the significance of MFPO from three aspects:
>
> **First**, complex multi-goal RL tasks require expressive policies to enable efficient exploration and multi-modal policy learning. As observed in AntMaze (Fig. 14 in https://anonymous.4open.science/r/MFPO-pics/BPTT.pdf), although SAC adopts the MaxEnt RL objective to encourage diversity, the unimodal nature of Gaussian policies limits it to a single behavior mode. This can lead to convergence to a single feasible solution or even failure to find valid solutions (e.g., SAC fails on AntMaze-v4). In contrast, diffusion-based policies can model complex multi-modal action distributions, enabling simultaneous exploration of multiple behavior modes.
>
> **Second**, diffusion-based policies incur substantially higher computational cost due to iterative sampling. As shown in Fig. 2 and Tab. 1, they require significantly more training (\~5×) and inference (\~10×) time than classical methods such as SAC and TD3. To address this limitation, MFPO adopts MeanFlow models, which enable high-quality action generation with only a few sampling steps, thereby substantially reducing training and inference time.
>
> **Third**, the MaxEnt RL objective is essential for effectively optimizing MeanFlow policies. As shown in Fig. 13 and Fig. 14 in https://anonymous.4open.science/r/MFPO-pics/BPTT.pdf, BPTT, a naive baseline that optimizes MeanFlow policies under the standard RL objective, consistently underperforms MFPO. On MuJoCo, it achieves lower returns, and on AntMaze, it collapses to a single solution, whereas MFPO discovers multiple feasible paths. This suggests that, despite their expressivity, MeanFlow models trained with standard RL objectives tend to produce near-deterministic or unimodal policies, as they focus on maximizing Q values. In contrast, MFPO incorporates the MaxEnt objective to explicitly promote exploration and multi-modal policy learning.
>
> Therefore, combining MeanFlow models with the MaxEnt RL framework is both natural and necessary to achieve efficient exploration, effective multi-modal policy learning, and improved computational efficiency, which are not jointly achieved by prior methods.
>
> **Q1: Same as W1.1**
>
> **Q2: Same as W2**
>
>
> [1] Sferrazza, Carmelo, et al. Humanoidbench: Simulated humanoid benchmark for whole-body locomotion and manipulation. https://arxiv.org/abs/2403.10506
>
> [2] Li, Zechu, et al. Learning multimodal behaviors from scratch with diffusion policy gradient. NeurIPS 2024.

---

> > ### Author Rebuttal · Reviewer_iTvu · 2026-04-01
> >
> > I have reviewed your experimental results on AntMaze. The visualizations demonstrate the multi-modal advantages of using flow matching. Your explanation regarding the improved learning efficiency of flow matching has addressed several of my initial concerns.
> >
> > I also noted the reported loss and variance (of the trace estimator), which, in my view, demonstrate that the proposed method is stable and convergent in this environment. While the empirical evidence is solid, the submission would be even stronger with a theoretical proof of convergence or a specific strategy that guarantees it.
> >
> > Personally, I find the AntMaze experiments insightful. I have decided to increase my score.

---

> > > ### Author Response · Authors · 2026-04-02
> > >
> > > Here we provide the theoretical analysis of MFPO, and we will add it in the revised manuscript.
> > >
> > > **Soft Policy Evaluation**
> > >
> > > Assuming that ADN loss (Eq. 17) and soft Bellman error (Eq. 29) are minimized, the learned Q-network recovers the true soft Q-function of the current policy.
> > >
> > > *Proof (sketch)*: The Skilling–Hutchinson trace estimator provides an unbiased estimate of the instantaneous divergence. Therefore, by MeanFlow theory, minimizing ADN loss enables ADN to approximate the true action likelihood. In this condition, minimizing the soft Bellman error ensures that the Q-network converges to the true soft Q-function of the current policy.
> > >
> > > **Soft Policy Improvement**
> > >
> > > Let $K_1$ and $K_2$ denote the sample sizes of two SNIS estimators, and $\Pi$ denote the class of parameterized MeanFlow policies. As $K_1, K_2 \to \infty$, and assuming sufficient network capacity such that $\pi^\*(a \mid s) \propto \exp(Q^{\pi_{\text{old}}}(s,a)/\alpha) \in \Pi$, minimizing the policy loss (Eq. 28) results in the new policy $\pi_{\text{new}} = \pi^\*(a \mid s)$, which improves upon $\pi_{\text{old}}$.
> > >
> > > *Proof (sketch)*: From the bias analysis of AIVE in App. B.4, as $K_1, K_2 \to \infty$, AIVE provides an unbiased estimator of the instantaneous velocity field corresponding to $\pi^\*(a \mid s)$. By MeanFlow theory, minimizing the MeanFlow loss (Eq. 28) recovers the target policy $\pi^*(a \mid s)$. Since $\pi^\*(a \mid s)$ is the minimizer of Eq. 4, it follows that $\pi_{\text{new}}=\pi^\*(a \mid s)$ improves upon $\pi_{\text{old}}$.
> > >
> > > **Conclusion**
> > >
> > > According to soft policy iteration (Sec. 3.2), by repeatedly alternating between soft policy evaluation and soft policy improvement, MFPO will converge to the optimal MaxEnt policy within $\Pi$.

---

### Official Review · Reviewer_if1s · 2026-03-08

**Soundness:** 2
**Presentation:** 3
**Significance:** 2
**Originality:** 2
**Overall Recommendation:** 4
**Confidence:** 3

**Summary:**

This paper addresses the computational inefficiency of diffusion-based policy representations in online reinforcement learning, which require many iterative sampling steps during both training and inference. The authors propose Mean Flow Policy Optimization (MFPO), which replaces standard flow-matching policies with a MeanFlow model.

**Compliance With Llm Reviewing Policy:**

Affirmed.

**Final Justification:**

The author has addressed most of my concerns. But it still remains to explain how entropy-regularized RL + flow matching can outperform diffusion policy, which is usually for action diversity. So I am moving my score to "weak accept".

**Key Questions For Authors:**

1. **On the efficiency-expressiveness trade-off**: You position MFPO as striking a balance between expressiveness and efficiency. Have you considered or tested simpler multi-modal alternatives (e.g., mixture of Gaussians, normalizing flows) that might achieve a similar trade-off with less machinery?
2. **On the average divergence network**: How sensitive is the final performance to the accuracy of the divergence network? If you artificially degrade the divergence estimates (e.g., by training the divergence network less frequently), how quickly does performance deteriorate?
3. **On the relationship to consistency models**: Consistency models also aim for a few-step generation. Could MFPO's approach (average divergence network + adaptive velocity estimation) be applied to consistency-model-based policies? What structural property of MeanFlow specifically enables your approach?

**Limitations:**

yes

**Strengths And Weaknesses:**

# Strength

1. **Integration with MaxEnt RL**. Rather than heuristically combining MeanFlow with RL (e.g., just replacing the policy network), the authors carefully address the two fundamental challenges: likelihood estimation and policy improvement. The average divergence network and adaptive velocity estimation are technically sound solutions that follow naturally from MeanFlow theory.
2.  **Efficiency improvement with empirical support**. The paper convincingly demonstrates that MFPO reduces training time by ~50% and inference time by ~3-4x compared to most diffusion baselines (Table 1, Figure 2f, Figure 6g) while maintaining competitive performance. The wall-clock comparisons are particularly valuable and often missing from prior work.

# Weakness
1. **Limited novelty in individual components**. The MeanFlow model is directly adopted from Geng et al. (2025). The SNIS combination with ESS weighting is a standard importance sampling technique. The average divergence network follows by straightforward analogy from the MeanFlow identity. While the integration is non-trivial, no single component represents a substantial methodological advance. What is the author's insight into moving MeanFlow models to policy optimization? Do we have something special to pay attention to?
2. **The policy improvement step through meanflow matching lacks theoretical justification**. The paper acknowledges (lines 310-313 of Sec. 4.4) that "the minimizer of this objective generally does not coincide with that of the KL divergence in Eq. (4) when the Boltzmann distribution lies outside the policy class," but then dismisses this concern by assuming "the solutions are close in practice due to the expressiveness of the MeanFlow policy class." This is hand-waving and very confusing for me.
3. **Lack of theoretical guarantees or convergence analysis**. The paper provides no formal analysis of whether MFPO converges to the optimal MaxEnt policy, what the approximation error of the average divergence network introduces, or how the bias of the SNIS estimators affects policy optimization.
4. **Benchmark scope is narrow**. The evaluation is limited to MuJoCo locomotion (5 tasks) and DMC (6 tasks), all of which are relatively low-dimensional continuous control tasks. Tasks with higher-dimensional action spaces, sparse rewards, or where multi-modality is truly critical for diffusion policies but are absent.

---

> ### Author Rebuttal · Authors · 2026-03-31
>
> **W1: Novelty of individual components and their intergration.**
>
> **A1**: Please refer to our response to **W1** of reviewer 1SiM.
>
> **W2: Theoretical analysis of the policy improvement step.**
>
> **A2**: Soft Policy Improvement: let $K_1$ and $K_2$ denote the sample sizes of two SNIS estimators, and $\Pi$ denote the class of parameterized MeanFlow policies. As $K_1, K_2 \to \infty$, and assuming sufficient network capacity such that $\pi^\*(a \mid s) \propto \exp(Q^{\pi_{\text{old}}}(s,a)/\alpha)$ lies in $\Pi$, minimizing the policy loss (Eq. 28) results in the new policy $\pi_{\text{new}} = \pi^*(a \mid s)$, which improves upon $\pi_{\text{old}}$.
>
> Proof (sketch): From the bias analysis of AIVE in App. B.4, as $K_1, K_2 \to \infty$, AIVE provides an unbiased estimator of the instantaneous velocity field corresponding to $\pi^\*(a \mid s)$. By MeanFlow theory, minimizing the MeanFlow loss (Eq. 28) recovers the target policy $\pi^*(a \mid s)$. Since $\pi^\*(a \mid s)$ is the minimizer of Eq. 4, it follows that $\pi_{\text{new}}=\pi^\*(a \mid s)$ improves upon $\pi_{\text{old}}$.
>
> **W3.1: Convergence analysis of MFPO.**
>
> **A3.1**: Soft policy Evaluation: assuming that ADN loss (Eq. 17) and soft Bellman error (Eq. 29) are minimized, the Q-network recovers the true soft Q-function of the current policy.
>
> Proof (sketch): The Skilling–Hutchinson trace estimator provides an unbiased estimate of the instantaneous divergence. Therefore, by MeanFlow theory, minimizing ADN loss enables ADN to approximate the true action likelihood. In this condition, minimizing the soft Bellman error ensures that the Q-network converges to the true soft Q-function of the current policy.
>
> According to soft policy iteration (Sec. 3.2), by repeatedly alternating between soft policy evaluation and soft policy improvement, MFPO will converge to the optimal MaxEnt policy.
>
> **W3.2: Approximation error of ADN.**
>
> **A3.2**: Please refer to our response to **Q1** of reviewer yqZ7.
>
> **W3.3: Bias of the SNIS estimator.**
>
> **A3.3:** Increasing sample sizes in SNIS reduces both bias and variance of the estimator. We provide learning curves with different sample sizes in Fig. 8(a) in App. E.4. Increasing sample sizes beyond the default setting ($K_1=16, K_2=32$) yields only marginal performance gains, suggesting that the estimation bias is already small in practice and has limited impact on policy optimization.
>
> **W4: Multimodal policy optimization and higher-dimensional tasks.**
>
> **A4**: Please refer to our response to **W2** of reviewer 1SiM.
>
> **Q1: Comparison with GMM and normalizing flows.**
>
> **A5**: We compare MFPO with GMM (SAC-GMM, implemented ourselves) and normalizing flows (MEow [1]) in Fig. 10 of https://anonymous.4open.science/r/MFPO-pics/GMM_NF.pdf. MFPO outperforms both SAC-GMM and MEow.
>
> For GMM, selecting the appropriate number of components is task-dependent and non-trivial in practice. For normalizing flows, the requirement of invertible transformation constrains the class of network parameterization, which may limit their expressivity. These factors may contribute to the observed performance gap compared to MFPO.
>
> **Q2: An analysis for the training frequency of ADN.**
>
> **A6**: We evaluate different update frequencies $f$ for ADN, as shown in Fig. 12 of https://anonymous.4open.science/r/MFPO-pics/ADN.pdf. Reducing the update frequency to $f = 1/8$ leads to noticeable performance degradation, as an under-trained ADN produces inaccurate action likelihood estimates that can interfere with policy optimization. In contrast, frequencies with $f \geq 1/2$ consistently achieve strong performance, indicating that ADN is sufficiently trained under these settings.
>
> **Q3: Relationship to consistency models (CM).**
>
> **A7**: CM maps a point $x_t$ on an ODE trajectory to its starting point via $x_0 = f(x_t, t)$. This can be interpreted as modeling the average velocity field $u(x_t, 0, t)$ over $[0,t]$, since $x_0 = x_t - t\cdot u(x_t, 0, t)$.
>
> In contrast, MeanFlow models generalize this formulation by modeling the average velocity field over an arbitrary time interval $[r, t]$. This generalization is critical for compatibility with the MaxEnt RL framework:
>
> (1) Action likelihood evaluation: Training ADN requires access to the instantaneous velocity field. In MeanFlow, setting $r = t$ reduces the average velocity to the instantaneous velocity, while CM lacks access to the instantaneous velocity.
>
> (2) Soft policy improvement: The training target of MeanFlow models is constructed by an unbiased estimator of the instantaneous velocity, which can be estimated via SNIS. In contrast, the training target of CM $x_0 = f(x_t, t)$ involves an integral over the instantaneous velocity, which is intractable.
>
> Therefore, in terms of compatibility with the MaxEnt RL framework, MeanFlow models are more suitable as policy representations than CM.
>
> [1] Chao, Chen-Hao, et al. Maximum entropy reinforcement learning via energy-based normalizing flow. NeurIPS 2024.

---

> > ### Author Rebuttal · Reviewer_if1s · 2026-04-02
> >
> > I appreciate the authors' efforts in conducting the additional experiments, and the new empirical results are indeed impressive. However, my primary concern regarding the novelty of the individual components and the theoretical connection between them remains unaddressed.
> >
> > MFPO is a highly complex framework comprising multiple moving parts, such as a mean-flow model for the policy, a divergence network, and a velocity estimation module (as noted in the response to Reviewer 1SiM). The fundamental weakness of this work is the lack of a **clear, unifying rationale** for why these specific modules are combined. Currently, the methodology reads like a collection of disparate new techniques stacked together and suddenly outperforming other baselines, rather than a cohesive architecture driven by a principled, underlying mechanism. In particular, why does MFPO outperform BPTT? Should we give credit to the MaxEnt RL objective? If so, why can MaxEnt RL + meanflow policy give us such an improvement while MaxEnt RL itself not? Given that MaxEnt RL can help, should we directly consider an entropy-regularized generative model for the policy, like the Schrödinger bridge?
> >
> > I am happy to increase my score if the author can justify the connection and pinpoint their insight from such a complicated framework.

---

> > > ### Author Response · Authors · 2026-04-02
> > >
> > > We would like to emphasize that **learning expressive policies via MeanFlow under the MaxEnt RL framework is both fundamentally important and largely unexplored in prior work**. Existing methods typically make a trade-off: classical approaches (e.g., SAC) adopt the MaxEnt objective but rely on simple unimodal policies, limiting exploration and expressivity, while recent generative approaches (e.g., diffusion/flow-based policies) improve expressivity but lack a principled integration with MaxEnt RL for efficient exploration and policy learning. **To our knowledge, MFPO is the first work that unifies MeanFlow policy parameterization with MaxEnt RL into a coherent framework**, enabling **efficient exploration, effective multi-modal policy learning, and tractable policy optimization simultaneously**.
> > >
> > > With this in mind, we clarify that MFPO is not a collection of ad-hoc modules, but a **principled framework derived from a unified objective: optimizing expressive (MeanFlow) policies under the MaxEnt RL framework**. Each component is introduced to address a specific challenge arising from this objective, and they are **functionally coupled rather than arbitrarily combined**.
> > >
> > > **First, why MeanFlow + MaxEnt RL?**
> > >
> > > Complex multi-goal tasks require expressive policies to support efficient exploration and multi-modal policy learning. As observed in AntMaze (Fig. 14 in https://anonymous.4open.science/r/MFPO-pics/BPTT.pdf), although SAC adopts the MaxEnt RL objective to encourage diversity, the unimodal nature of Gaussian policies limits it to a single behavior mode. This can lead to convergence to a single feasible solution or even failure to find valid solutions (e.g., SAC fails on AntMaze-v4). In contrast, diffusion-based policies can model complex multi-modal action distributions, enabling simultaneous exploration of multiple behavior modes. However, **expressivity alone is insufficient**: without entropy regularization, BPTT collapse toward near-deterministic solutions due to pure Q maximization (the optimal policy under the standard RL objective is $\pi(a\mid s)=\arg\max_{a}Q(s,a)$). Therefore, BPTT may suffer from **insufficient exploration** and be trapped in local minimum. This explains why **MaxEnt RL, which explicitly promote exploration and multi-modal policy learning, is essential**, and why **MaxEnt RL + expressive policy (MeanFlow)** yields gains beyond either component alone.
> > >
> > > **Second, why these specific components?**
> > >
> > > Once we adopt the objective of optimizing expressive (MeanFlow) policies under the MaxEnt RL framework, three key challenges arise:
> > >
> > >  (1) **Efficient action generation**. Diffusion-based policies incur substantially higher computational cost due to iterative sampling. To address this limitation, MFPO adopts MeanFlow models, which enable high-quality action generation with only a few sampling steps, thereby significantly reducing training and inference time.
> > >
> > >  (2) **Tractable likelihood estimation**. Soft policy evaluation requires computing the action likelihood of the MeanFlow policy, which involves an intractable integral over the divergence of the instantaneous velocity field (Eq. 12). We propose to train the Average Divergence Network, which enables efficient and accurate likelihood estimation.
> > >
> > >  (3) **Effective soft policy improvement**. When approximating the Boltzmann distribution of the Q-function, previous method (MaxEntDP [1]) relies on SNIS with a Gaussian proposal, which becomes inefficient at large $t$. We instead introduce an additional proposal (the current policy) and combine the two SNIS estimators using ESS as weights, yielding stable and accurate instantaneous velocity estimation across all $t$.
> > >
> > > These components are therefore **not optional add-ons**, but **minimal and necessary components** of a coherent optimization pipeline for MeanFlow policies.
> > >
> > > **Third, complexity and overhead.**
> > >
> > > We test the training overhead and performance gain of each component, as shown in Tab. 4 of https://anonymous.4open.science/r/MFPO-pics/overhead.pdf. Importantly, these components do not make the method practically complex: they introduce only a small computational overhead, while yielding substantial performance gains.
> > >
> > > **Finally, why not alternatives such as Schrödinger Bridge?**
> > >
> > > By modeling the average velocity field, MeanFlow mitigates discretization error in few-step sampling, achieving strong one-step generation performance. This makes it a **natural choice for policy parameterization when both expressivity and fast sampling are required**.
> > >
> > > Nevertheless, MeanFlow is not the only viable policy class for realizing the MaxEnt RL objective. As the reviewer notes, alternatives such as Schrödinger Bridge are also promising directions. We view them as **complementary approaches** and leave their exploration to future work.
> > >
> > > [1] Dong, X., Cheng, J., Zhang, X. S. Maximum entropy reinforcement learning with diffusion policy. ICML 2025.

---

### Official Review · Reviewer_1SiM · 2026-03-09

**Soundness:** 2
**Presentation:** 2
**Significance:** 3
**Originality:** 2
**Overall Recommendation:** 4
**Confidence:** 2

**Summary:**

In general, this paper draws inspiration from existing methods that use diffusion models to represent policies. The authors found that these methods can incur significant training and inference overhead during iterative RL optimization. To mitigate this issue, the authors do not use a diffusion model but instead employ meanflow as the policy model, improving the performance of diffusion-based RL algorithms. The authors identified and addressed two main challenges of using meanflow policies: action likelihood evaluation and soft policy improvement. Experiments were then conducted on benchmarks such as Mujoco and DeepMind Control Suite. The results demonstrate that the proposed MFPO (i.e., meanflow policy optimization) method achieves similar or better performance while reducing the time and cost of training and inference.

**Compliance With Llm Reviewing Policy:**

Affirmed.

**Final Justification:**

During the rebuttal, the author has addressed my original concerns through supplementary experiments and detailed analysis. Consequently, I find the rebuttal to be good and have decided to maintain my original positive rating.

**Key Questions For Authors:**

Most of the issues are mentioned in the weakness section, so please see Weakness and:

* Could the authors elaborate on the settings for sampling steps T in more detail, such as how to set sampling steps for different tasks and how to adaptively select them? How sensitive is this parameter?
* Furthermore, it would be very helpful if the authors could further discuss the runtime costs of each component.

**Limitations:**

The authors discuss the limitations in the paper and believe that fewer sampling steps remain a direction worth exploring. I think the authors could further discuss the failure cases of the proposed method, which might help readers understand in which situations the proposed method is unusable.

**Strengths And Weaknesses:**

## Strengths

From the perspective of using Meanflow to parameterize policy models, this paper is novel and well-motivated, as diffusion models indeed require a significant number of sampling and training time. Methods with fewer sampling steps can indeed reduce the time cost of training and testing.

The paper and appendices also provide an analysis of the proposed method, and extensive experiments validate its effectiveness. Therefore, I believe the method is reliable to a certain extent. In short, the experiments, including numerous tasks and ablation studies on MujoCo and DeepMind Control Suite, allow readers to better understand the role of each component in the proposed method. The discussion and analysis related to likelihood estimation are indeed key to applying Meanflow.

Furthermore, the paper demonstrates a reduction in training time, which is truly significant. The comparative baseline is also quite comprehensive.

## Weaknesses

This paper appears to attempt to combine several existing works/components. While there are many experiments, this paper could further discuss some unique innovations.

Furthermore, I believe the motivation for MeanFlow as a policy model to replace the diffusion model is sufficient. However, if the authors could discuss or attempt applications to higher-dimensional tasks, such as multimodal fine-tuning or multimodal policy optimization (any discussion is welcome), it might make the proposed method more readily applicable to readers.

Although the authors discuss the overall computational cost,  this paper still lack sufficient analysis of the computational cost of each auxiliary component. A brief discussion of these aspects by the authors would be highly beneficial.

---

> ### Author Rebuttal · Authors · 2026-03-31
>
> **W1: Unique innovations of the proposed method.**
>
> **A1**: MFPO introduces three key components: (1) adopting MeanFlow models as policy representations, (2) the Average Divergence Network (ADN), and (3) Adaptive Instantaneous Velocity Estimation (AIVE). These components are all essential for achieving high and stable performance.
>
> To highlight the necessity of our design, we further develop BPTT, a naive baseline that optimizes MeanFlow policies under the standard RL objective. Specifically, BPTT learns the standard Q-function and updates the policy by backpropagating Q-gradients through the flow rollout.
>
> As shown in Fig. 13 and Fig. 14 of https://anonymous.4open.science/r/MFPO-pics/BPTT.pdf, BPTT consistently underperforms MFPO. On MuJoCo benchmarks, it achieves lower returns, while on AntMaze tasks, it is limited to a single solution, whereas MFPO discovers multiple feasible paths. This is because, despite their expressive capacity, MeanFlow models trained with standard RL objectives tend to collapse to near-deterministic or unimodal policies, as they are driven to maximize the Q values.
>
> In contrast, MFPO incorporates the MaxEnt RL objective to explicitly encourage exploration and multi-modal policy learning, which is crucial for fully leveraging the representational power of MeanFlow models. In addition, ADN enables efficient and accurate likelihood estimation, and AIVE facilitates effective soft policy improvement, which are both critical for optimizing MeanFlow policies under the MaxEnt RL objective.
>
> **W2: Multimodal policy optimization and higher-dimensional tasks.**
>
> **A2**: We evaluate MFPO and competing methods on AntMaze [1] (multimodal policy optimization tasks) and HumanoidBench [2] (with a 61-dimensional action space), with results shown in Fig. 14 of https://anonymous.4open.science/r/MFPO-pics/BPTT.pdf and Fig. 15 of https://anonymous.4open.science/r/MFPO-pics/HumanoidBench.pdf.
>
> On AntMaze, MFPO successfully explores multiple feasible paths, whereas BPTT and SAC are limited to a single mode. On HumanoidBench, MFPO consistently outperforms SAC, FlowRL, and DIME across all tasks, demonstrating its effectiveness in high-dimensional control tasks.
>
> **W3: The computational cost of each auxiliary component.**
>
> **A3**: We remove the auxiliary components of MFPO, and add them one by one to test the training overhead and performance gain of each component, showing the results in Tab. 4 of https://anonymous.4open.science/r/MFPO-pics/overhead.pdf. The proposed components incur only a small computational overhead, yet yield substantial performance improvements.
>
> **Q1: The settings of sampling steps T.**
>
> **A4**: We provide training curves for different sampling steps $T$ in Fig. 8(a) in App. E.4. We find that $T=2$ performs well across all tested tasks and is therefore used as the default setting. In practice, we recommend starting with $T=2$ for new tasks and gradually increasing it if further performance improvements are needed.
>
> **Q2: Same as W3.**
>
> **Limitations: The failure cases of the proposed method.**
>
> **A5**: We observe that using two sampling steps achieves strong performance across the tested benchmarks, whereas reducing the number of steps to one leads to a noticeable performance degradation. As a result, the current method is not well-suited for real-time control tasks that require one-step action generation. In future work, we plan to investigate more advanced policy optimization and generative modeling techniques to further reduce the number of sampling steps.
>
> [1] Li, Zechu, et al. Learning multimodal behaviors from scratch with diffusion policy gradient. NeurIPS 2024.
>
> [2] Sferrazza, Carmelo, et al. Humanoidbench: Simulated humanoid benchmark for whole-body locomotion and manipulation. https://arxiv.org/abs/2403.10506

---

> > ### Author Rebuttal · Reviewer_1SiM · 2026-04-01
> >
> > The author has addressed my original concerns through supplementary experiments and detailed analysis. Consequently, I find the rebuttal to be good and have decided to maintain my original positive rating.

---

### Official Review · Reviewer_yqZ7 · 2026-03-13

**Soundness:** 2
**Presentation:** 2
**Significance:** 2
**Originality:** 2
**Overall Recommendation:** 2
**Confidence:** 5

**Summary:**

The paper introduces Mean Flow Policy Optimization (MFPO), a reinforcement learning algorithm that leverages MeanFlow models to represent policies in continuous control tasks. The primary motivation is to overcome the high computational training and inference costs associated with multi-step diffusion models in RL while maintaining their expressive, multi-modal generation capabilities. To integrate MeanFlow into the Maximum Entropy (MaxEnt) RL framework, the authors propose two main technical contributions: an Average Divergence Network (ADN) to approximate action likelihoods for soft policy evaluation , and an adaptive instantaneous velocity estimation method utilizing Self-Normalized Importance Sampling (SNIS) for policy improvement. The method is evaluated on standard MuJoCo and DeepMind Control Suite benchmarks against several diffusion and traditional RL baselines.

**Compliance With Llm Reviewing Policy:**

Affirmed.

**Final Justification:**

The authors have addressed most of my questions during the discussion phase, and I appreciate their efforts in providing clarifications. However, I still have significant concerns regarding the level of methodological novelty, which is central to my evaluation. As these concerns remain unresolved, I will maintain my original score and assessment.

**Key Questions For Authors:**

**ADN Error Accumulation**: The Average Divergence Network relies on an unbiased trace estimator that is then fitted via an L2 loss. How do you account for the approximation error of $\delta_\omega$ compounding within the soft policy iteration loop? Can you provide a clearer step-by-step intuition of how this network practically accelerates sampling without destabilizing the Q-value targets?

**Efficiency Claims vs. FlowRL**: Table 1 indicates that FlowRL achieves lower inference latency (0.42 ms vs. 0.46 ms) and requires fewer sampling steps (1 vs. 2) compared to MFPO. Given this, how do you justify the claim that MFPO addresses the efficiency bottleneck better than existing state-of-the-art flow-based RL methods?

**Marginal Performance Gains**: In Figure 2, the episodic returns for MFPO plateau at roughly the same level as MaxEntDP and DIME on several environments (e.g., Walker2d-v3, Ant-v3). Where exactly does the advantage of multi-modal exploration, claimed in the introduction, manifest in these specific continuous control tasks if the final returns are nearly identical?

**Missing Relevant Baselines**: The current experimental setup does not include comparisons with highly relevant flow-based and one-step baselines, specifically One-step Meanflow[1] and SAC-Flow[2]. Since this work directly tackles the efficiency and performance of flow-based generative policies, omitting these baselines weakens the empirical claims. Can you provide a direct comparison (in terms of performance, sample efficiency, and inference latency) against One-step Meanflow and SAC-Flow?

[1] One-Step Flow Policy Mirror Descent. https://arxiv.org/pdf/2507.23675

[2] SAC Flow: Sample-Efficient Reinforcement Learning of Flow-Based Policies via Velocity-Reparameterized Sequential Modeling. ICLR2026.

**Limitations:**

While the authors briefly mention that MFPO still requires two sampling steps and propose reducing it to one step as future work, they have not adequately discussed the potential failure modes of the Average Divergence Network. Furthermore, the limitations surrounding the approximation errors introduced by the SNIS estimator in highly skewed reward landscapes are not discussed. Finally, the authors fail to address the limitation of their empirical evaluation regarding the exclusion of key related baselines like One-step Meanflow and SAC-Flow. I suggest adding a dedicated paragraph critically analyzing the computational overhead and stability trade-offs of training the auxiliary divergence network, as well as a more comprehensive discussion on how MFPO positions itself against a broader set of baseline methods.

**Strengths And Weaknesses:**

**Strengths**:

*Originality*: Applying the recently proposed MeanFlow models  to continuous control RL is a novel and interesting perspective. Shifting the learning target from instantaneous velocity to average velocity to reduce discretization errors  is a mathematically creative way to address the sampling bottleneck in diffusion-based policies.

*Soundness (Motivation)*: The authors correctly identify a major bottleneck in current diffusion-based RL literature: the high computational overhead of iterative generation.

**Weaknesses**:

*Presentation & Soundness (Average Divergence Network Logic)*: The introduction and execution of the Average Divergence Network (ADN) is mathematically dense but logically convoluted. The paper proposes $\delta_\omega$ to approximate the time-averaged divergence to avoid explicit Jacobian computation. However, the narrative fails to clearly explain how this auxiliary network interacts with the primary policy network during the highly dynamic early stages of RL training without suffering from compounding approximation errors. The leap from the continuous time identity in Eq. 36  to the empirical training target in Eq. 40 using the Skilling-Hutchinson trace estimator  is difficult to follow and lacks a clear, intuitive justification for why this specific, highly complex architecture is necessary for efficient sampling.

*Significance (Performance Improvements)*: The experimental results do not demonstrate a compelling or significant performance leap over existing state-of-the-art baselines. Looking at the MuJoCo learning curves in Figure 2, MFPO performs comparably to baselines like DIME and MaxEntDP, but the asymptotic returns are not strictly superior. For instance, in Ant-v3 and Walker2d-v3, the final performance overlaps heavily with the variance of the baselines. The performance gains are too marginal to justify the significant added complexity of the ADN and adaptive SNIS machinery.

*Significance & Soundness (Efficiency Claims)*: The paper repeatedly claims that MFPO "considerably reduces training and inference time". However, the evidence is not sufficiently convincing when compared to concurrent efficient flow-matching works. According to Table 1, MFPO requires 2 sampling steps and achieves an inference time of 0.46 ms. In contrast, the FlowRL baseline requires only 1 sampling step and achieves a faster inference time of 0.42 ms. While MFPO is significantly faster than older 16-20 step methods like DIME or DACER, it does not establish a new state-of-the-art in efficiency, which undermines the core motivation of the paper.

---

> ### Author Rebuttal · Authors · 2026-03-31
>
> **W1: The motivation and training method of ADN.**
>
> **A1**: Soft policy evaluation requires computing the action likelihood of the MeanFlow policy, which involves an intractable integral over the divergence of the instantaneous velocity field (Eq. 12). To enable efficient and accurate likelihood estimation, we introduce the Average Divergence Network (ADN) to approximate the ratio between the divergence integral and the time interval (Sec. 4.2).
>
> Training ADN follows a similar paradigm to MeanFlow models and relies on an unbiased estimator of the instantaneous divergence. Specifically, we adopt the Skilling–Hutchinson trace estimator, which is both unbiased and computationally efficient. The full training procedure of ADN, the likelihood-aware sampling process, and the complete MFPO algorithm are detailed in Alg. 1–3 in App. C.
>
> **W2: Performance improvement over baselines.**
>
> **A2**: Our goal is to reduce the training and inference cost of diffusion/flow-based policies without sacrificing performance. As shown in Fig. 2, Fig. 6, and Tab. 1, MFPO achieves ~2× faster training and ~3–4× faster inference compared to diffusion-based baselines, while maintaining comparable or superior performance.
>
> **W3: Efficiency comparison with FlowRL.**
>
> **A3**: FlowRL adopts a midpoint ODE solver, where each sampling step requires two forward network evaluations (line 281). As a result, the number of forward evaluations (NFE) for action sampling, the primary factor determining training and inference speed, is the same for FlowRL and MFPO.
>
> The slightly lower inference time reported for FlowRL (0.42 ms vs. 0.46 ms for MFPO) is due to not using the action selection technique; when this technique is enabled, the inference speed of the two methods becomes nearly the same.
>
> Moreover, FlowRL exhibits inferior performance on both MuJoCo and DMControl benchmarks, whereas MFPO consistently maintains strong performance across these tasks.
>
> **Q1: The effect of approximation error of ADN.**
>
> **A4**: In Fig. 11 of https://anonymous.4open.science/r/MFPO-pics/ADN.pdf, we report the training loss of ADN, the variance of the Skilling–Hutchinson trace estimator, and the relative error of action likelihood estimation between ADN and a 1000-step Euler ODE solver (used as a near ground-truth reference). The results show that ADN closely matches the likelihood estimates of the 1000-step Euler solver while requiring only two steps.
>
> We further visualize the action likelihood estimation of ADN on a toy 2D task (App. E.1). Despite minor approximation errors, ADN preserves the relative structure of the likelihood landscape, consistently assigning higher values to high-likelihood action regions. This property allows the estimated likelihood to serve as an intrinsic reward that encourages exploration of less-visited action regions.
>
> Therefore, the approximation error is small and does not adversely affect the policy optimization process.
>
> **Q2: Same as W3**
>
> **Q3: Comparison with MaxEntDP and DIME.**
>
> **A5**: MaxEntDP, DIME, and MFPO are all diffusion-based MaxEnt RL methods, and therefore enable multi-modal exploration, outperforming unimodal Gaussian policies. The key advantage of MFPO lies in its efficiency: it significantly reduces both training and inference time compared to MaxEntDP and DIME, while maintaining strong performance.
>
> **Q4: Adding more baselines.**
>
> **A6**: One-step MeanFlow (FPMD-M) performs policy improvement by weighting the MeanFlow loss on the old policy with the exponential Q-function, resulting in $\pi_{k+1} \propto \pi_k \cdot  \exp(Q/\tau)$. However, this update is overly conservative in online RL, leading to slower policy improvement.
>
> SAC-Flow interprets flow-based policies as sequential models and parameterizes the velocity network with sequence architectures (e.g., GRU or Transformer) to stabilize training. In stead of back-propagating Q gradiant through the flow rollout as in SAC-Flow, MFPO directly computes the regression target of the average velocity network, enabling stable training without specialized architectural design.
>
> We compare these methods in Fig. 9 and Tab. 3 of https://anonymous.4open.science/r/MFPO-pics/SACFlow_FPMD.pdf. MFPO outperforms both SAC-Flow and One-step MeanFlow on most tasks. It also requires less training and inference time than SAC-Flow. While One-step MeanFlow is slightly faster to train, it suffers from a significant performance drop.
>
> **Limitations:** All limitations have been addressed above except for the following three:
>
> **L1: Bias of the SNIS estimator.**
>
> **A7**: Please refer to our response to **W3.3** of reviewer if1S.
>
> **L2: The computional overhead and performance gain of auxillary components**
>
> **A8:** Please refer to our response to **W3** of reviewer 1SiM.
>
> **L3: Positioning of MFPO against a broader set of baselines.**
>
> **A9**: Please refer to our response to **W3** of reviewer iTvu.

---

> > ### Author Rebuttal · Reviewer_yqZ7 · 2026-04-03
> >
> > Thank you for your patient response and I have some more questions: Is it possible to compare the algorithm with SAC-Flow under the same environment? Maybe it would be better as a flow-based baseline. Thank you.

---

> > > ### Author Response · Authors · 2026-04-03
> > >
> > > As discussed in our earlier response (A6), we have already included a comparison with SAC-Flow and One-step MeanFlow under the same environments. As shown in Fig. 9 and Tab. 3 of https://anonymous.4open.science/r/MFPO-pics/SACFlow_FPMD.pdf, MFPO consistently outperforms SAC-Flow on most tasks, while also requiring less training and inference time. These results demonstrate that MFPO provides both improved performance and better computational efficiency compared to this flow-based baseline.
> > >
> > > We agree that SAC-Flow and One-step MeanFlow are important related methods, and we will include them in the main paper and report the corresponding comparison in the revised manuscript.

---

### Decision · Program_Chairs · 2026-04-30

**Decision:**

Accept (regular)

**Comment:**

The paper addresses the issue of many iterations in diffusion models by extending mean flows to policy optimization. This allows taking only two steps saving training time while maintaining strong performance.

The reviewers generally appreciated the work and in favor of accepting the paper. The rejecting reviewer's originally raised points have been substantively addressed in rebuttal. During discussion, it was clear that the remaining concerns are not significant.

As such I recommend accepting the paper.